# Disruption of riboflavin biosynthesis in mycobacteria establishes riboflavin pathway intermediates as key precursors of MAIT cell agonists

Melissa D. Chengalroyen[1☉], Nurudeen Oketade[2☉], Aneta Worley[3], Megan Lucas[2], Luisa Maria Nieto Ramirez[2], Mabule L. Raphela[1], Gwendolyn M. Swarbrick[3], Paul S. Soma[2], Mandisa Zuma[1], Digby F. Warner[1,4], Deborah A. Lewinsohn[5], Carolina Mehaffy[2], Erin J. Adams[6], William Hildebrand[7], Karen M. Dobos[2*], Valerie Mizrahi[1,4*], David M. Lewinsohn[3,8*]

1 UCT Molecular Mycobacteriology Research Unit, Institute of Infectious Disease and Molecular Medicine & Department of Pathology, University of Cape Town, Cape Town, South Africa, 2 Department of Microbiology, Immunology and Pathology, Colorado State University, Fort Collins, Colorado, United States of America, 3 Department of Pulmonary, Allergy & Critical Care Medicine, Oregon Health and Science University, Portland, Oregon, United States of America, 4 Wellcome Centre for Infectious Disease Research in Africa, University of Cape Town, Cape Town, South Africa, 5 Division of Infectious Diseases, Oregon Health and Science University, Portland, Oregon, United States of America, 6 Department of Biochemistry and Molecular Biology, University of Chicago, Chicago, Illinois, United States of America, 7 Department of Microbiology and Immunology, University of Oklahoma Health Sciences Centre, Oklahoma City, Oklahoma, United States of America, 8 Portland VA Medical Center, Portland, Oregon, United States of America

☉ These authors contributed equally to this work.
* karen.dobos@colostate.edu (KD); valerie.mizrahi@uct.ac.za (VM); lewinsod@ohsu.edu (DML)

## Abstract

Mucosal-associated invariant T (MAIT) cells exhibit an intrinsic ability to recognize and respond to microbial infections. The semi-invariant antigen recognition receptor of MAIT cells specifically detects the non-polymorphic antigen-presenting molecule, major histocompatibility complex class I-related protein 1 (MR1), which primarily binds riboflavin-derived metabolites of microbial origin. To further interrogate the dependence of these antigens on riboflavin biosynthesis in mycobacteria, we deleted individual genes in the riboflavin biosynthesis pathways in *Mycobacterium smegmatis* (Msm) and *Mycobacterium tuberculosis* (Mtb) and evaluated the impact thereof on MAIT cell activation. Blocking the early steps of the pathway by deletion of RibA2 or RibG profoundly reduced, but did not completely ablate, MAIT cell activation by Msm or Mtb, whereas deletion of RibC, which catalyzes the last step in the pathway, had no significant effect. Interestingly, deletion of the lumazine synthase (RibH) specifically enhanced MAIT cell recognition of Mtb whereas loss of lumazine synthase activity had no impact on MAIT cell activation by Msm. MAIT cell activation by Msm was likewise unaffected by blocking the production of the MAIT cell antagonist, $F_o$ (by inhibiting its conversion from the riboflavin pathway intermediate,

**Data availability statement:** The data that support the findings of this study will be are publicly available from Dryad after acceptance. The data can be assessed by reviewers with the links provided in the manuscript. Upon acceptance, the data and can be accessed with the identifiers, DOI: 10.5061/dryad.vq83bk454, DOI: 10.5061/dryad.4tmpg4fnv, and DOI: 10.5061/dryad.z08kprrr1.

**Funding:** This research was supported by a Harry Oppenheimer Fellowship from the Oppenheimer Memorial Trust (to V.M.; www.omt.org.za), and grants from the Bill and Melinda Gates Foundation INV-004757 (to V.M.; www.gatesfoundation.org), the Broad Institute (to V.M.; www.broadinstitute.org), the South African Medical Research Council (to V.M.; www.samrc.ac.za), the Department of Science and Innovation and National Research Foundation of South Africa (to V.M.; www.nrf.ac.za), the National Institutes of Health (NIH; www.niaid.nih.gov) R01AI147954 (to E.J.A., W.H. and D.M.L., MPIs), and AWD100279 (to K.M.D. and C.M.). The funders had no role in study design, data collection and analysis, decision to publish, or preparation of the manuscript.

**Competing interests:** The authors have declared that no competing interests exist.

5-amino-6-D-ribitylaminouracil (5-A-RU), through the deletion of *fbiC*). Together, these results confirm a central role for 5-A-RU in generating mycobacterial MR1 ligands and reveal similarities and differences between Msm and Mtb in terms of the impact of riboflavin pathway disruption on MAIT cell activation.

## Author summary

Mucosal-associated invariant T (MAIT) cells are an abundant population of innate-like T-cells in humans which respond to microbial infections. These specialized cells recognize the MR1 molecule, which presents microbial metabolites derived from riboflavin (vitamin B2) biosynthesis. These cells are enriched in the airways, and, in some cases, reduced in the peripheral blood of tuberculosis (TB) infected individuals suggestive of a role in the early response to infection by *Mycobacterium tuberculosis*. In this study, we investigated the effect of deleting individual genes in the riboflavin biosynthesis pathway on MAIT cell activation by *Mycobacterium tuberculosis* or *Mycobacterium smegmatis*. Our findings revealed that disrupting early stages in the pathway profoundly reduced, but did not eliminate MAIT cell activation by both mycobacterial species. However, blocking the penultimate step in the pathway, catalyzed by the lumazine synthase, RibH, led specifically to increased MAIT cell recognition of Mtb. Our results confirm the pivotal role of the riboflavin pathway intermediate, 5-A-RU, in generating mycobacterial ligands that serve as MAIT cell agonists. By enhancing our understanding of how MAIT cells recognize mycobacterial infections, the results of this study could inform strategies for the development of vaccines and/or immunotherapies for TB.

## Introduction

MR1-restricted T cells (MR1Ts) are characterized by their dependence on the highly conserved molecule, MR1 [1–4]. A subset of MR1Ts, Mucosal Associated Invariant T cells (MAITs), have been characterized by their usage of a semi-invariant T cell receptor (TRAV1–2) [1,5], their expression of the transcription factor PLZF, and expression of CD161 and CD26 [6]. While these cells are among the most abundant in the human circulation and in mucosal sites, their function was largely unknown until 2010 when they were found to recognize a variety of microbes including the major human pathogen, *Mycobacterium tuberculosis* (Mtb) [7,8]. A seminal discovery was made in 2012 when Kjer-Nielsen *et al*. [3] demonstrated that MR1 could present the metabolite, 5-(2-oxopropylideneamino)-6-D-ribitylaminouracil (5-OP-RU), derived from an intermediate in the riboflavin biosynthesis pathway of *Salmonella* sp., to MAIT cells. These authors argued that microbes producing riboflavin are uniquely poised to activate these cells. Microbially derived MR1 ligands and antigens have since been found to be more diverse than originally described [4]; others that have

been discovered include DMRL, an intermediate of the riboflavin pathway, and photolumazines I, III and V, which were identified in *Mycobacterium smegmatis* (Msm) and are also derived from the riboflavin pathway [9,10]. However, the source of MR1 ligands that are stimulatory for MAIT cells may extend beyond the riboflavin pathway as evidenced by our finding that the TRAV1–2-negative, TRAV12–2-expressing T-cell clone, D462-E4, recognizes ligands derived from *Streptococcus pyogenes*, a riboflavin auxotroph [11]. This suggests that MR1Ts can distinguish between microbes in a T-cell receptor (TCR)-dependent manner.

The central role of the microbial riboflavin pathway in MAIT cell activation has been confirmed in different organisms. These include deletion mutants of *Lactococcus lactis* [4], *Salmonella* Typhimurium [4,12], *Escherichia coli* [13,14], *Streptococcus pneumoniae* [15], *Klebsiella pneumoniae* [16] and *Aspergillus fumigatus* [17] rendered auxotrophic for riboflavin as well as natural variants of *S.* Typhimurium [18] and *S. pneumoniae* [15] with altered expression of the riboflavin pathway. Moreover, engineered overexpression of specific riboflavin pathway genes in Msm, Mtb or *M. bovis* BCG was shown to enhance MAIT cell activation, albeit differentially between the three mycobacterial species and to varying extents, depending on the gene targeted for over-expression [19]. Building on our prior work demonstrating the essentiality of riboflavin biosynthesis in Mtb and Msm, and the ability of both organisms to transport and assimilate riboflavin [20], we set out to investigate the impact of riboflavin pathway disruption on MAIT cell activation by these mycobacteria. We generated mutants of Mtb and Msm auxotrophic for riboflavin by knocking out individual genes in the biosynthesis pathway and analyzed the impact thereof on the expression of the pathway genes and proteins, the abundance of pathway intermediates, and on MAIT cell activation. We observed a complete loss of DMRL when RibA2 was deleted in Msm and Mtb and when RibG was deleted in Msm. We also observed in Msm that the lumazine synthases, RibH1 and RibH2, were capable of synthesizing DMRL, while Mtb relied on a single lumazine synthase for DMRL biosynthesis. We show that blocking the early steps of the pathway by deleting either RibA2 or RibG profoundly reduced but did not completely eliminate MAIT cell activation whereas deletion of RibC, which catalyzes the last step in the pathway, had no discernable effect on MAIT cell activation by Msm or Mtb. Moreover, deletion of RibH specifically enhanced MAIT cell recognition of Mtb whereas loss of lumazine synthase activity had no significant impact on MAIT cell activation by Msm. MAIT cell activation by Msm was likewise unaffected by blocking the production of the MAIT cell antagonist, $F_0$, which is derived from the riboflavin pathway intermediate, 5-A-RU, by deletion of *fbiC*. Our results confirm a central role for 5-A-RU in the production of mycobacterial MR1 ligands and the non-dominant role of DMRL as an MR1 ligand in these organisms. Our results also reveal similarities and differences between Msm and Mtb in terms of the impact of riboflavin pathway disruption on MAIT cell activation. These findings will guide future studies in the field of MR1-dependent immune responses towards Mtb.

## Results

### Construction of deletion mutants in the riboflavin pathway of Msm and Mtb

In a recent study [20], we characterized the riboflavin biosynthesis pathway in Mtb H37Rv and Msm mc²155 and confirmed that it comprises a bifunctional GTP cyclohydrolase II/3,4-DHBP synthase (RibA2); riboflavin deaminase/5-amino-6-(5-phosphoribosylamino) uracil reductase (RibG); an unidentified phosphatase; lumazine synthase (RibH), and riboflavin synthase (RibC) (Fig 1A and 1B). In Msm and Mtb the *ribC*, *ribA2* and *ribH* genes are operonic [20], similar to the genomic organisation in *B. subtilis* where the riboflavin biosynthesis genes are arranged on a five-gene operon [21]. In *B. subtilis*, FMN riboswitches regulate the expression of riboflavin biosynthesis and transport [22]; however these regulatory elements have not been identified in the genomes of mycobacteria. Riboflavin is the precursor of flavin mononucleotide (FMN) and flavin adenine dinucleotide (FAD), which are produced by the bifunctional enzyme, RibF. The biosynthetic pathway also leads to the production of the deazaflavins, $F_0$ and $F_{420}$, from the pathway intermediate, 5-A-RU, via the sequential action of the enzymes FbiC, FbiA and FbiB [23]. Together, the flavin and deazaflavin cofactors enable the activity of a myriad of enzymes in mycobacteria, organisms which adopt 'flavin-intensive' lifestyles [24,25].

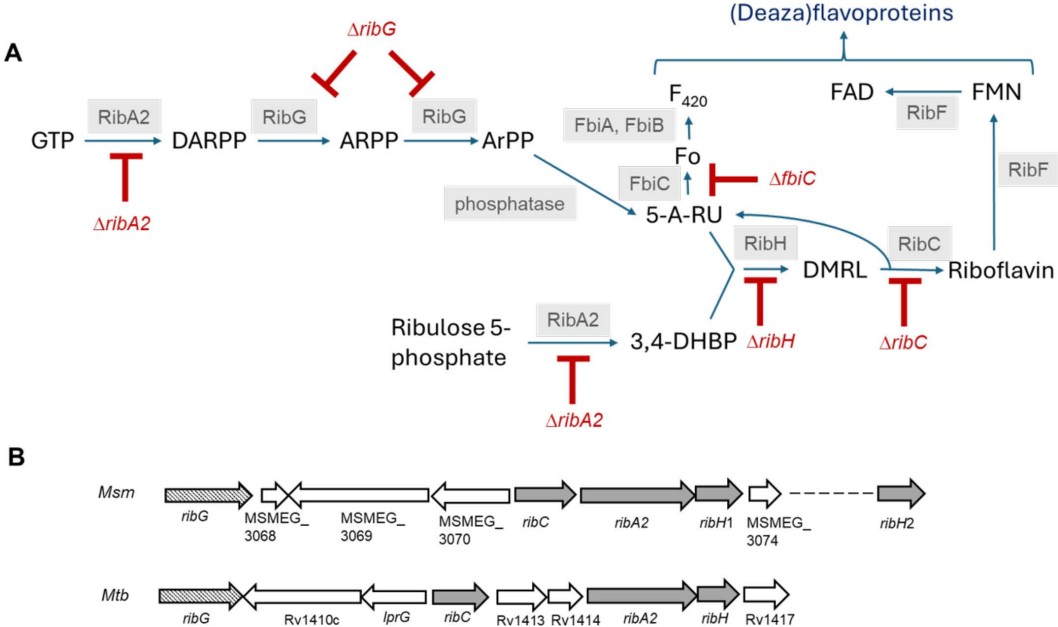

**Fig 1. Riboflavin biosynthesis and utilization pathways in mycobacteria. (A)** In the biosynthesis pathway, GTP and ribulose-5-P are converted to 2,5-diamino-6-(5-phospho-D-ribosylamino)-pyrimidin-4(3H)-one (DARPP) and 3,4-dihydroxyl-2-butanone 4-phosphate (3,4-DHBP), respectively, by the bifunctional GTP cyclohydrolase II/ DHBP synthase, RibA2. DARPP is deaminated and the side chain subsequently reduced by the bifunctional riboflavin deaminase/ 5-amino-6-(5-phosphoribosylamino) uracil reductase, RibG, to form 5-amino-6-ribitylamino-2,4(1H,3H)-pyrimidinedione-5-phosphate (ArPP). Dephosphorylation of ArPP by an unknown phosphatase generates 5-amino-6-D-ribitylaminouracil (5-A-RU). 5-A-RU together with 3,4-DHBP are condensed by the lumazine synthase, RibH (RibH1/RibH2), to yield 6,7-dimethyl-8-ribityllumazine (DMRL). Two molecules of DMRL are converted to riboflavin and 5-A-RU by the riboflavin synthase, RibC via a dismutation reaction. In the utilization pathway, riboflavin is converted to FMN and FAD by the bifunctional kinase/ FAD synthetase, RibF. The deazaflavin, $F_{420}$, is produced from 5-A-RU by the sequential action of FbiC, FbiA and FbiB. The deletion mutations are shown in red. **(B)** Genomic context of the riboflavin biosynthesis genes in Msm and Mtb.

A notable difference between Msm and Mtb is the redundancy in lumazine synthase activity in Msm where this function is served by a canonical RibH (MSMEG_3073, designated herein as RibH1) – the homologue of the sole RibH in Mtb (Rv1416) – and a second homologue, MSMEG_6598, designated here as RibH2 [20]. In prior work, we used a panel of inducible CRISPRi hypomorphs in genes involved in riboflavin biosynthesis and utilization to confirm the essentiality of the riboflavin pathway for mycobacterial growth and survival *in vitro.* We also demonstrated that Mtb and Msm can assimilate exogenous riboflavin but have yet to establish whether this occurs through passive diffusion or facilitated transport via a non-canonical transporter/s [20]. However, residual expression of target genes in induced hypomorphs could confound the ability to differentiate the contributions of individual riboflavin pathway intermediates in MAIT cell recognition of mycobacteria. To avoid this potential complication, we created Msm and Mtb riboflavin auxotrophs carrying deletions in individual genes in the riboflavin biosynthesis pathway for the purpose of this study (S1 Fig). Importantly, the deletion mutations in *ribA2* eliminated both GTP cyclohydrolase II and DHBP synthase domains of RibA2; the former catalyzes the first step in the pathway whereas the latter produces a substrate for the RibH enzyme, which catalyzes the penultimate step.

The growth phenotypes of the Msm and Mtb mutants and their complemented counterparts were assessed in liquid and on solid media, with or without riboflavin supplement (Fig 2, S2 and S3 Figs). The Msm Δ*ribA2*, Δ*ribG*, Δ*ribH2*Δ*ribH1* and Δ*ribC* mutants were auxotrophic for riboflavin whereas the Msm Δ*fbiC* mutant showed no growth phenotype. When plating the wildtype and genetically complemented Msm auxotrophs on 7H10 agar with or without riboflavin supplementation, there was no significant difference in total CFU counts (S2B Fig). However, colonies of the complemented derivatives,

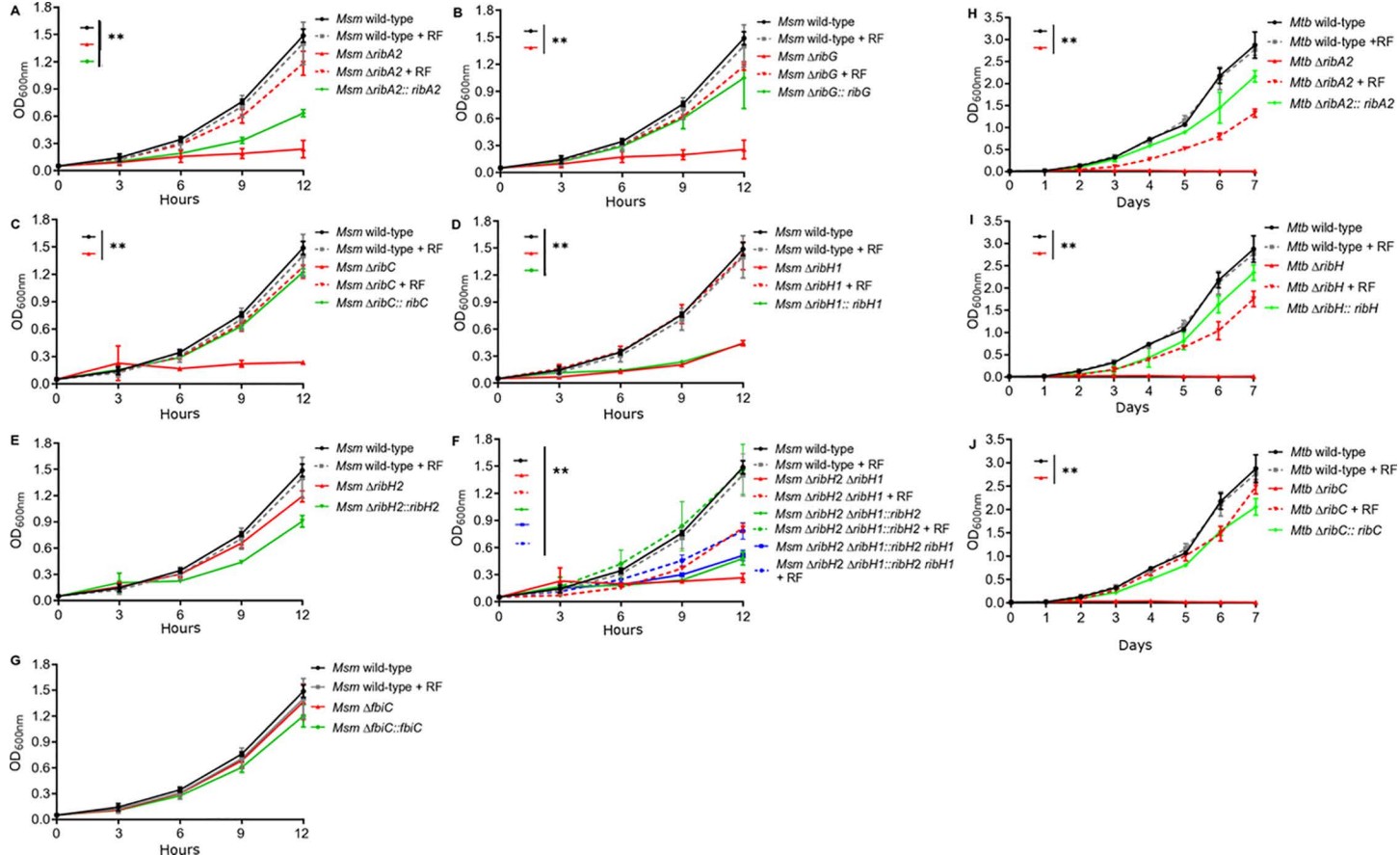

**Fig 2. Growth kinetics of wildtype, knockout and complemented mutant strains of Msm (A-G) and Mtb (H-J) in 7H9 media.** Growth of strains in liquid culture was monitored every 3 hours for 12 hours for Msm, and every day for 7 days for Mtb. Data are plotted as the mean, and error bars represent standard deviation from four biological replicates. Statistical comparisons against wildtype were performed at the 12-hr (for Msm) or 7-day (for Mtb) time point using area under the curve analysis (for Msm, panels A-G) or a one-way ANOVA and Dunnett's multiple comparison test (for Mtb, panels H-J) whereby statistical significance is represented by $p < 0.05$, $p < 0.001$ shown by *, ** respectively. Only statistically significant results ($p < 0.05$) are shown. RF; riboflavin.

Msm Δ*ribA2*::*ribA2,* Δ*ribH1*::*ribH1,* Δ*ribH2*Δ*ribH1*::*ribH2* and Δ*ribH2*Δ*ribH1*::*ribH2 ribH1* emerged later than the other strains (S2A Fig) and were significantly smaller than wildtype (S2C Fig). The retarded growth of these strains on solid media was consistent with their retarded growth kinetics in liquid culture (Fig 2A, 2D and 2F) which contrasts that other complemented derivatives for which restoration of growth to a level approximating the wildtype was observed (Fig 2B and 2C). This effect could be attributed to comparatively weak expression of the complementing genes, *ribA2* and *ribH1*, from nested promoters instead of the main operonic promoter located upstream of *ribC*. The ability of Msm Δ*ribH1* to grow in the absence of riboflavin supplement on solid media (S2A Fig) and in liquid culture (albeit significantly more slowly than wildtype; Fig 2D) confirmed the existence of an alternate source/s of lumazine synthase activity in Msm that can partly compensate for the loss of RibH1. The fact that the deletion of both *ribH1* and *ribH2* rendered Msm incapable of growth without riboflavin supplement on solid media (S2A Fig) or in liquid culture (Fig 2F) is consistent with the designation of RibH2 as an alternate lumazine synthase in Msm.

Mtb Δ*ribA2*, Δ*ribH* and Δ*ribC* were auxotrophic for riboflavin (S3A Fig). In all cases, growth of the Mtb knockout mutants with riboflavin, and of the complemented strains in liquid media was equivalent to wildtype (Fig 2H-2J).

Likewise, CFU counts of the knockout mutants grown on riboflavin, as well as of the complemented strains, were equivalent to wildtype (S3B Fig) although the colony size of the wildtype strain was on average smaller than the other strains (S3C Fig).

Riboflavin pathway transcript and protein levels in the Msm wildtype, knockout and complemented strains were measured by qRT-PCR and LC-MS/MS based Data Independent Acquisition (DIA) proteomics, respectively (Figs 3 and 4). As the auxotrophs required exogenous riboflavin for growth, we assessed the impact of riboflavin supplement on the levels of riboflavin pathway transcripts and proteins in wildtype Msm and found no significant effects. The absence of the cognate transcript and protein was confirmed in the knockout mutants with variable restoration thereof in the complemented derivatives in a manner consistent with the corresponding growth phenotypes (Fig 2 and S2 Fig). Certain knockouts also significantly affected the expression of other members of the pathway, e.g., a ~3-fold increase in *ribH2* transcript and concomitant increase in RibH2 protein was observed in the Msm Δ*ribH1* mutant relative to the wildtype suggesting a regulatory interdependence of expression between the two lumazine synthases (Fig 4C and 4D). However, the Δ*ribH1* mutant was still significantly growth-impaired relative to wildtype (Fig 2D) indicating that RibH2 cannot fully compensate for the loss of RibH1 in Msm, and suggesting that RibH1 is the main catalytic enzyme for production of riboflavin. The Δ*ribH2* Δ*ribH1*::*ribH2 ribH1* mutant also showed significantly retarded growth kinetics (Fig 2F), which improved to some extent in the presence of riboflavin albeit not to the level of the wildtype. Poor complementation of the Δ*ribH1* deletion mutation was evidenced by the markedly lower levels of *ribH1* transcript (~40-fold) and RibH1 protein (~3-fold) observed in the Δ*ribH1*::*ribH1* and Δ*ribH2*Δ*ribH1*::*ribH2 ribH1* mutants compared to the wildtype (Fig 4A and 4B), which could potentially explain the comparative growth impairment of these complemented strains (Fig 2D and 2F). Whole genome sequencing (WGS) revealed only two non-synonymous SNPs in Msm Δ*ribH2*Δ*ribH1*::*ribH2 ribH1* which were situated in a $\sigma^{70}$ family RNA polymerase sigma factor and an intergenic region, respectively. The contributions (if any) thereof to the growth phenotype of the complemented double mutant and incomplete growth restoration by exogenous riboflavin are unclear.

Statistically significant changes in transcript and/or protein levels were also observed in other Msm knockouts, specifically, *ribC* deletion was accompanied by a decrease in *ribG* transcript, RibG protein (Fig 3C and 3D) and RibA2 protein (Fig 3B), whereas *ribA2* deletion resulted in an increase in the levels of *ribH1* (Fig 4A) and *ribC* transcripts (Fig 3E) relative to wildtype in riboflavin-supplemented media. We also observed some differences in the effect of pathway gene knockout *vs.* knockdown on expression of other genes in the pathway, e.g., silencing of *ribH1* led to an upregulation of *ribG* and *ribF*, and silencing of *ribA2* suppressed the expression of *ribG*, *ribC* and *ribH1* in Msm, demonstrated in our previous study [20] but such effects were not observed by knockout of *ribH1* or *ribA2.* The differences may be due in part to the fact that the target proteins were only partially depleted in the knockdowns under the conditions of the expression analysis [20] but are entirely absent in the knockouts.

The Mtb Δ*ribA2*, Δ*ribH* and Δ*ribC* mutants likewise showed riboflavin auxotrophy with riboflavin-independent growth restored by genetic complementation (Fig 2H–2J and S3A Fig). Knockout and complementation of *ribA2, ribH* and *ribC* were further validated by transcript and protein quantification (Fig 5). Additionally, stable isotope-labeled standards were made for peptides derived from RibC and RibH with the goal of determining targeted peptide concentration and, by inference, a more accurate quantification of the intact protein using parallel reaction monitoring (PRM) proteomics (S6 and S7 Figs and S8 Table). Riboflavin supplementation reduced the expression of *ribC* and increased the expression of *ribA2* in wildtype Mtb (Fig 5A and 5E) but had no significant impact on expression of *ribH*. Also, the variable transcript levels had no impact on the abundance of their protein products (Fig 5B, 5D and 5F). Knockout of other pathway genes had no effect on RibA2 abundance (Fig 5B). RibH transcript and protein levels were markedly lower than wildtype in the complemented Mtb Δ*ribH* strain (Fig 5C and 5D) but still sufficient to restore growth to wildtype levels (Fig 2I). In contrast, *ribH* transcript and RibH protein levels were increased ~10–12-fold in the Δ*ribA2* mutant and its complement *vs.* wildtype (Fig 5C and 5D), a finding confirmed by absolute quantification of RibH using PRM.

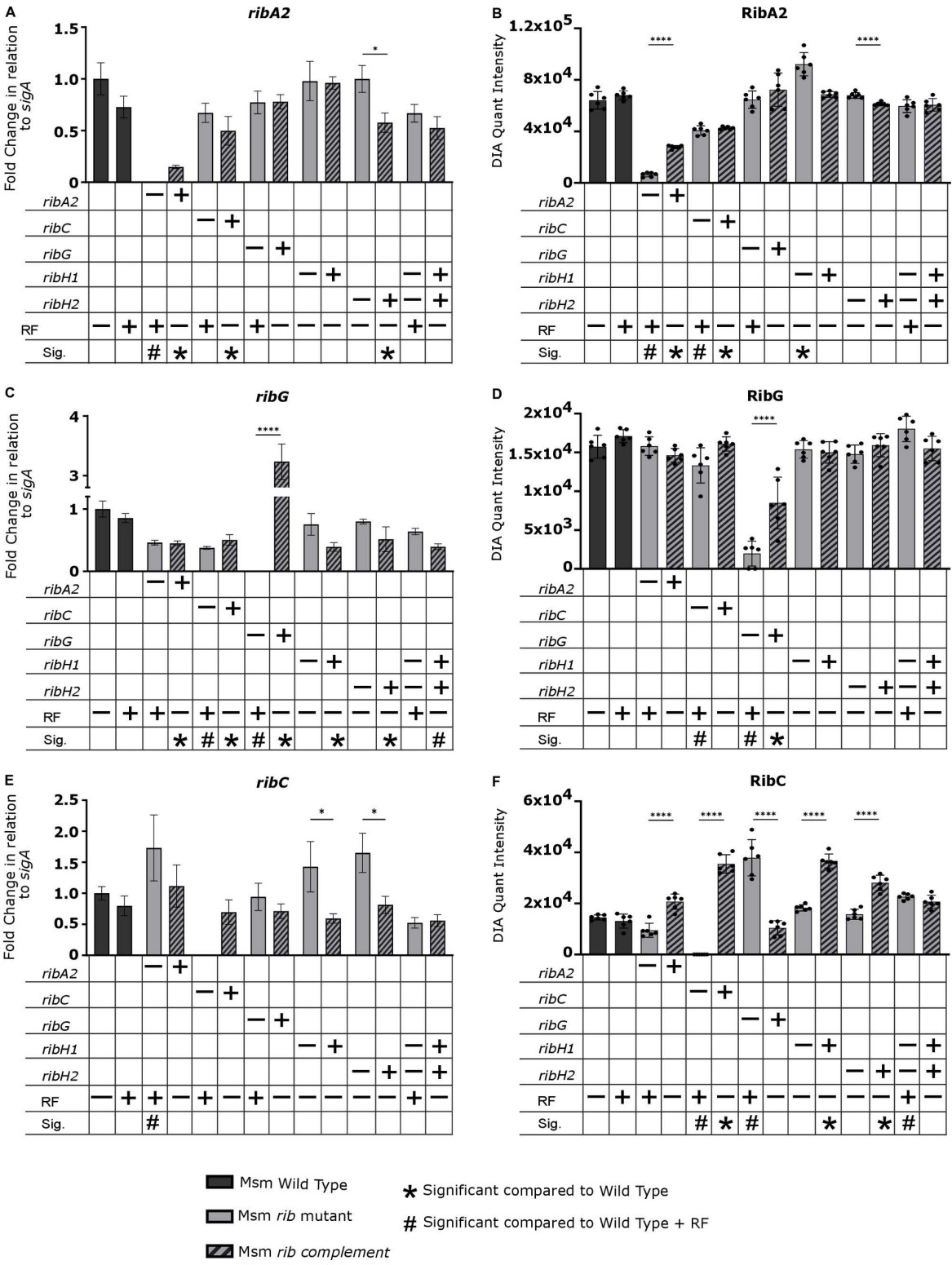

**Fig 3. Quantification of *ribA2*, *ribC* and *ribG* transcript and protein levels in Msm strains.** (**A**, **C**, and **E**) Fold changes in *sigA*-normalized transcript levels relative to wildtype. Transcript levels of the target genes were normalized to the housekeeping gene *sigA* (an essential housekeeping gene which is stably expressed) and scaled to the average of wildtype Msm to calculate the ΔΔCt and determine the fold difference in gene expression

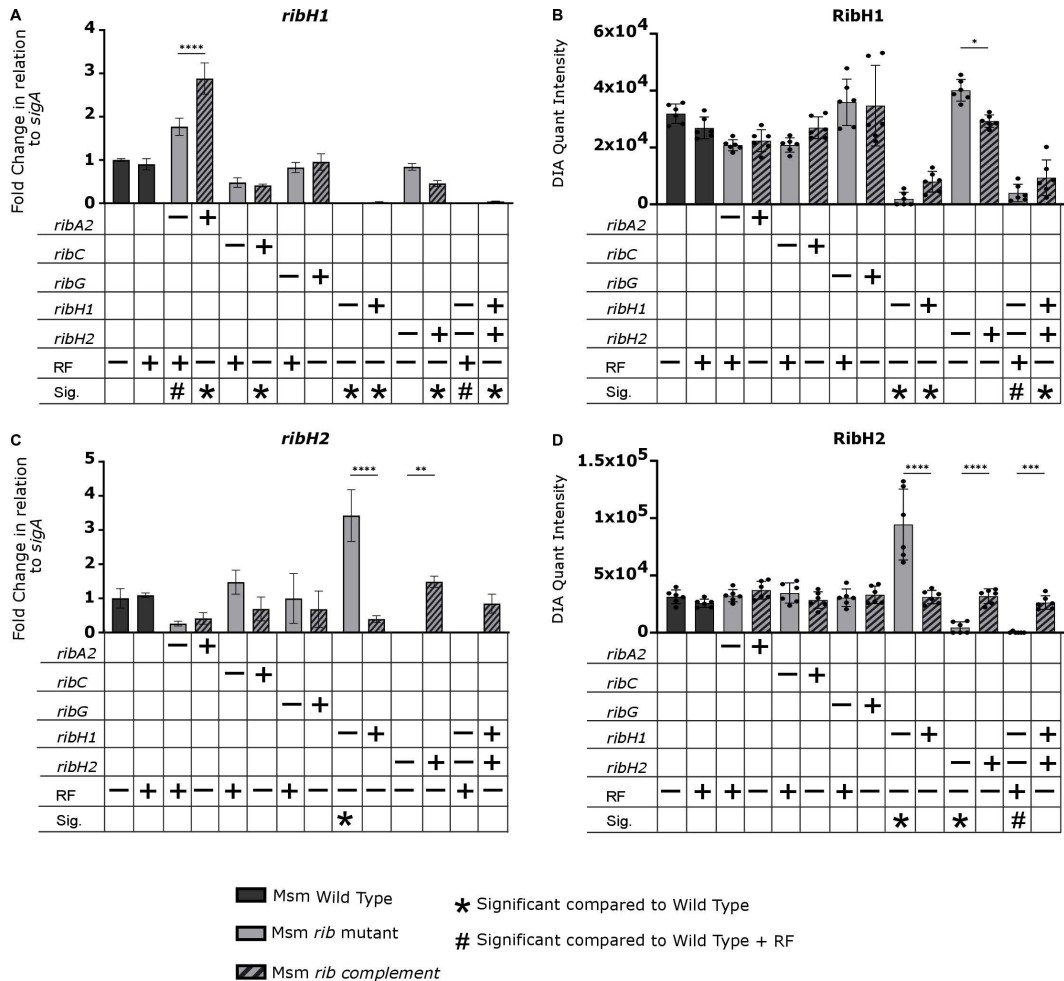

**Fig 4. Quantification of *ribH1*, *ribH2* transcript and protein levels of lumazine synthase genes in Msm strains. (A and C)** Fold changes in *sigA*-normalized transcript levels relative to wildtype. Transcript levels of the target genes were normalized to the housekeeping gene *sigA* (an essential housekeeping gene which is stably expressed) and scaled to the average of wildtype Msm to calculate the ΔΔCt and determine the fold difference in gene expression (n = 2 biological replicates, 2 technical replicates). **(B and D)** Protein abundance was measured using DIA proteomics (n = 3 biological replicates, 2 technical replicates). Data are shown as mean quantification ± SEM. Statistical comparisons were performed using a one-way ANOVA and Sidak's multiple comparison test whereby statistical significance is represented by p < 0.05, p < 0.001, p < 0.0005 p < 0.0001, shown by *, **, ***, **** respectively. Only statistically significant relationships are shown. RF, riboflavin. Significance in comparison to wildtype (+/- RF) is shown as symbols as described in legend (p < 0.05). – and + indicates deletion and complementation of the corresponding gene, respectively.

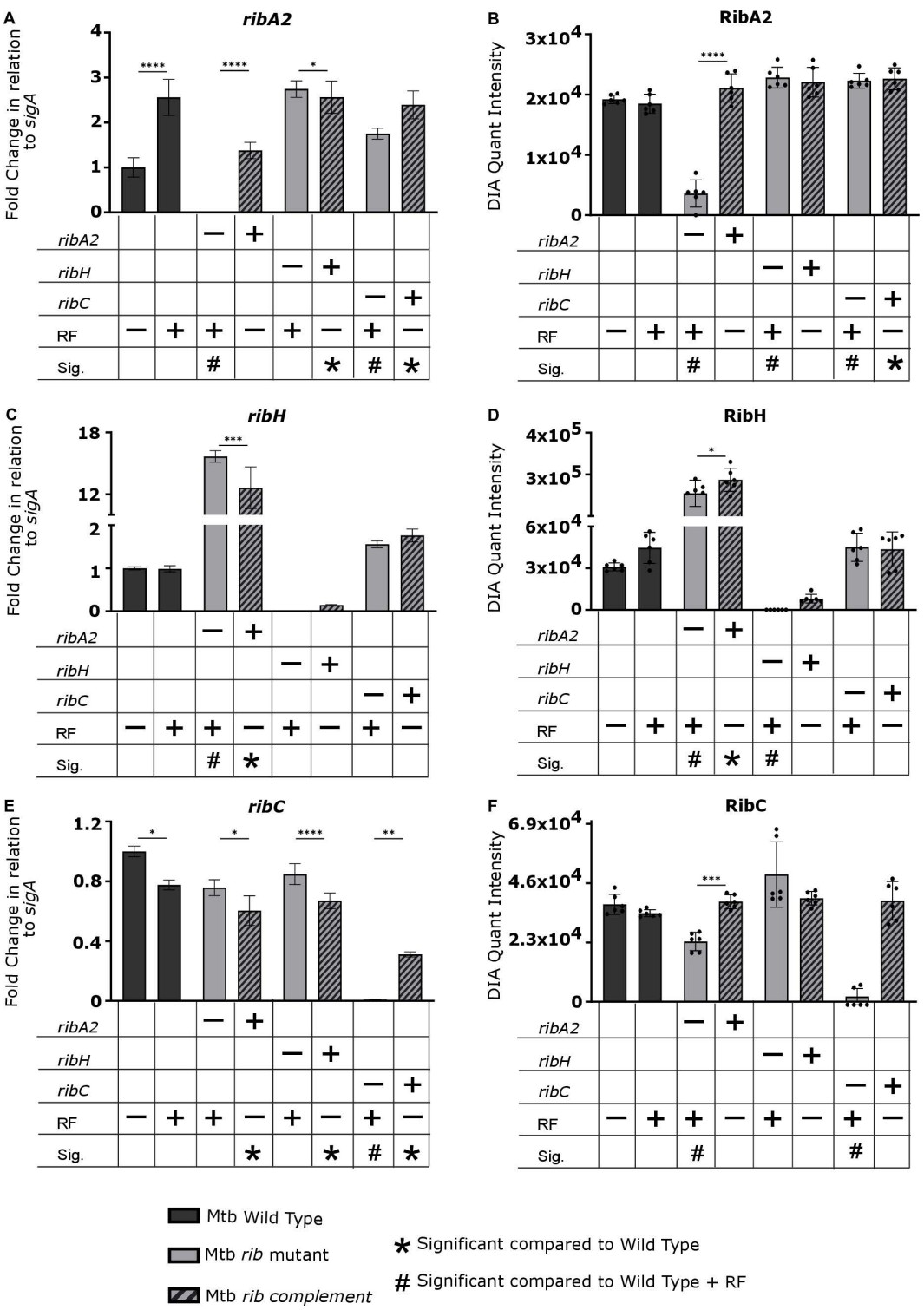

**Fig 5. Quantification of *ribA2*, *ribH* and *ribC* transcript and protein levels in Mtb strains. (A, C and E)** Fold changes in *sigA*-normalized transcript levels relative to wildtype. Transcript levels of the target genes were normalized to the housekeeping gene *sigA* (an essential housekeeping gene which is stably expressed) and scaled to the average of wildtype Mtb to calculate the ΔΔCt and determine the fold difference in gene expression (n = 3 biological replicates, 2 technical replicates). **(B and D)** Protein abundance was measured using DIA proteomics (n = 3 biological replicates, 2 technical

### RibH1 and RibH2 are capable of synthesizing DMRL in Msm while Mtb depends on a single lumazine synthase, RibH

Having confirmed the loss and restoration of gene expression and protein abundance in the cognate mutants and complemented strains of Msm, we next examined how these genetic disruptions affected metabolic intermediates within the pathway. To assess this, we measured the intracellular levels of 5-A-RU, DMRL, and riboflavin using LC-MS/MS Multiple Reaction Monitoring (MRM) metabolomics. DMRL was detected in all Msm strains except for the ΔribA2, ΔribG, and ΔribH1ΔribH2 mutants (Fig 6A and 6B). Although RibA2 protein expression was reduced in the ΔribA2::ribA2 complemented strain (Fig 3B), DMRL production was equivalent to wildtype levels (Fig 6A). We previously hypothesized that

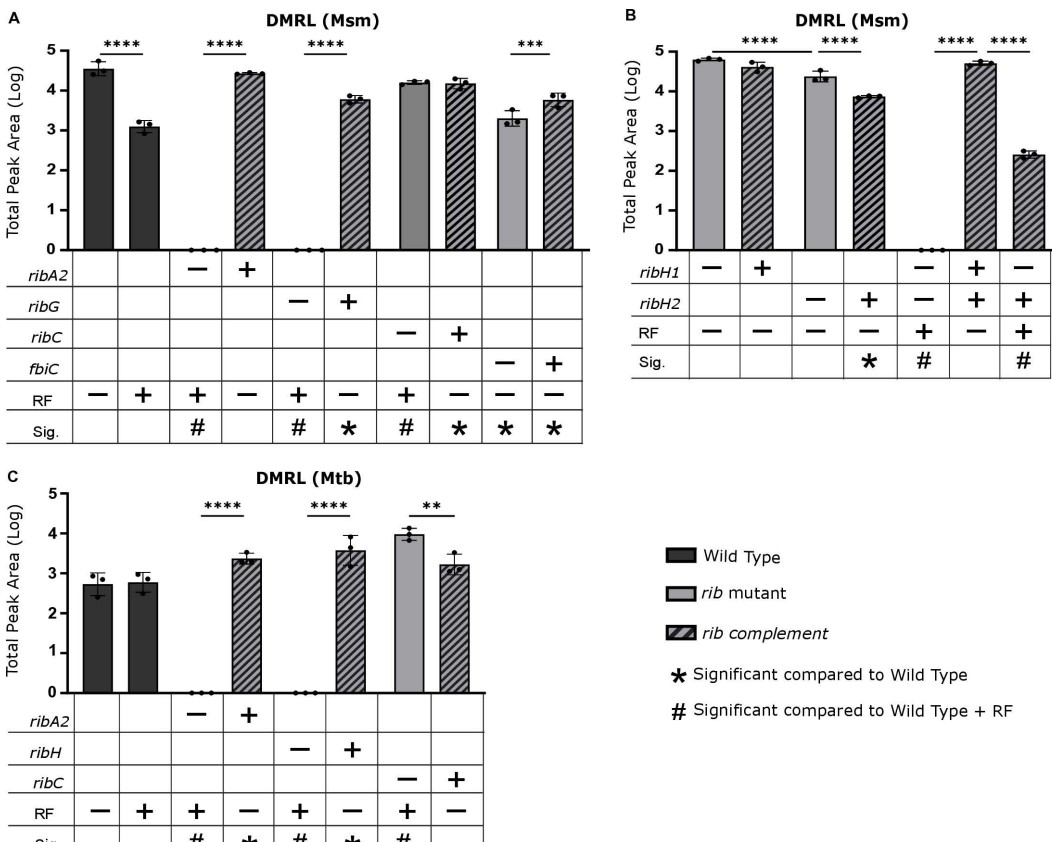

**Fig 6. Quantification of DMRL in Msm and Mtb strains.** Levels of intracellular DMRL were quantified in Msm (**A** and **B**) and Mtb (**C**) using MRM. Data are shown as mean quantification±SEM for three biological replicates. Quantification was carried out using chromatographic peak area of the most intense transition ion (m/z 327.1 → 193) of DMRL. Statistical comparisons were performed using a one-way ANOVA and Sidak's multiple comparison test whereby statistical significance is represented by $p < 0.05$, $p < 0.001$, $p < 0.0005$ $p < 0.0001$, shown by *, **, ***, **** respectively. Only statistically significant relationships are shown. RF, riboflavin; Sig., significance in comparison to wild type (+/- RF) is shown as symbols, as described in the legend ($p < 0.05$).

RibH1 and RibH2 may both independently produce DMRL, as supported by our observation that the deletion of *ribH1* did not completely abolish Msm viability (S2A Fig). Consistent with this hypothesis, we found that DMRL levels in individual *ribH1* and *ribH2* mutants were similar to those in wildtype, indicating both lumazine synthases can synthesize DMRL utilized for riboflavin biosynthesis (Fig 6B and S7A Fig). However, when comparing DMRL levels between lumazine synthase mutants, the *ribH2* mutant exhibited significantly lower levels compared to the *ribH1* mutant, and these reduced levels were not restored upon complementation (Fig 6B). In Mtb, deletion of *ribA2* and *ribH* resulted in the complete loss of DMRL production (Fig 6C, S9A Fig), reflecting the reliance of Mtb on a single lumazine synthase for DMRL synthesis. This observation aligns with our initial finding that deletion of *ribH* severely compromises the viability of Mtb. Additionally, significantly elevated DMRL levels observed in the *ribC* mutant suggest accumulation due to stalled riboflavin synthesis, a phenomenon not observed in Msm (Fig 6C). Riboflavin was consistently detected in the wildtype strain and in all complemented strains for both Msm and Mtb (S7B, S8 and S9B Figs). Since the riboflavin auxotroph mutants were grown in riboflavin, the substrate was also detected in those strains. For Msm, riboflavin levels in complemented strains typically mirrored wildtype levels, except for the Δ*ribC*::*ribC* and Δ*fbiC*::*fbiC* complemented strains, which exhibited slightly higher riboflavin levels (S8A Fig). Riboflavin quantification in the riboflavin prototrophic Δ*ribH1* and Δ*ribH2* mutants grown without riboflavin supplementation revealed that deletions of the lumazine synthase paralogs exerted distinct effects on riboflavin production: a minor albeit significant reduction in riboflavin production was observed in the Δ*ribH1* mutant, whereas riboflavin production in the Δ*ribH2* mutant was indistinguishable from wildtype (S8B Fig). In contrast, Mtb strains showed no significant differences in riboflavin levels between the wildtype and complemented strains (S8C Fig).

Despite including 5-A-RU in our MRM assay, we could not detect this intermediate in any tested strain, likely due to its highly labile nature, as noted during assay development (S10 Fig) and supported by published data [26]. However, we can infer from our proteomic and metabolic analysis that the loss of DMRL when *ribA2* and *ribG* were deleted is due to the loss of the substrate, 5-A-RU, required for its synthesis.

## Deletion of *ribA2* or *ribG* selectively ablates MAIT cell activation by Msm

To assess the impact of disrupting the riboflavin biosynthesis pathway on MAIT cell activation, Msm cultures grown in the presence of riboflavin were used to assess the ability to activate a panel of three MR1 restricted T cell clones: D426-G11, D481-C7 and D481-F12, TRAV1–2 + MAIT cell clones with varied TCRα chains (S1 Table) [9]. These were selected to reflect the diversity of ligands seen by MR1Ts with D481-C7 uniquely able to detect the photolumazine antigens derived from Msm [9]. Each clone was tested by IFN-γ ELISPOT for its ability to recognize dendritic cells infected with mycobacteria at an MOI of 10 (Fig 7). Deletion of *ribA2* or *ribG* resulted in an 82–99% reduction in T cell recognition of Msm for all clones tested, a phenotype reversed by complementation. Since we still observed minor MAIT cell activation for both the *ribA2* and *ribG* Msm deletion strains, we evaluated if this residual activity was MR1-dependent. We observed that the residual MAIT cell activation against both strains was blocked by an anti-MR1 antibody (S11 Fig). In contrast, deletion of *ribC* had no significant impact on T cell activation by Msm (Fig 7A, 7C and 7E). Deletion of *ribH1* and *ribH2* alone or in combination had no impact on recognition of Msm by the three TRAV1–2 + clones, D426-G11, D481-C7 and D481-F12.

In addition to its role as a substrate of RibH and precursor of DMRL, 5-A-RU is also required for the production of the deazaflavin MAIT cell antagonist, $F_0$ [23], via the action of FbiC. We therefore assessed the effect of blocking this route of 5-A-RU metabolism on MAIT cell activation using a *fbiC* deletion mutant of Msm. The Δ*fbiC* mutation had no effect on Msm viability (Fig 2) or recognition by the TRAV1–2 + T cell clones, D426-G11, D481-F12 and D481-C7 (Fig 7A, 7C and 7E).

## Divergent effects of *ribA2* vs. *ribH* deletion on MAIT cell activation by Mtb

The Mtb strains were tested for their ability to activate two of the MR1 restricted T cell clones described above, D426-G11, D481-C7, and D481-F12 (Fig 8). Each clone was tested by IFN-γ ELISPOT for its ability to recognize the dendritic cells infected with Mtb at an MOI of 10. As the strains varied in their ability to activate the MAIT cell clones, we normalized

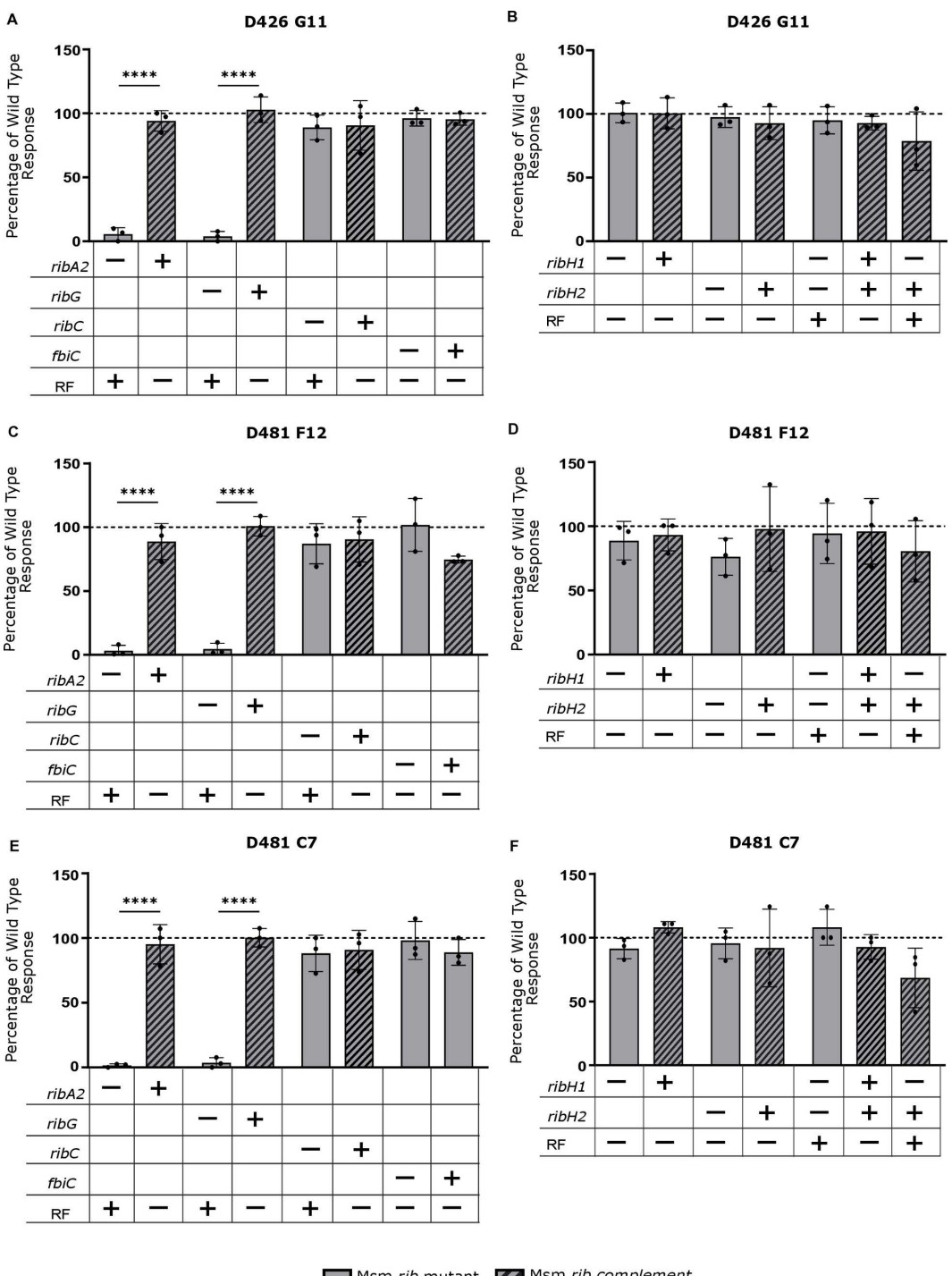

**Fig 7. Impact of riboflavin pathway mutations on MR1T recognition of Msm.** MR1T cell clone (1e4) IFN-γ response to dendritic cells (1e4) incubated with Msm at a MOI of 10. Response was normalized to wildtype Msm. All strains were grown with riboflavin. Wildtype Msm and positive control (PHA) response are shown in S12 Fig. Data are representative of n = 3 independent experiments. Statistical comparisons were performed using a one-way ANOVA and Sidak's multiple comparison test whereby statistical significance is represented by p < 0.05, p < 0.001, p < 0.0005 p < 0.0001, shown by *, **, ***, **** respectively.

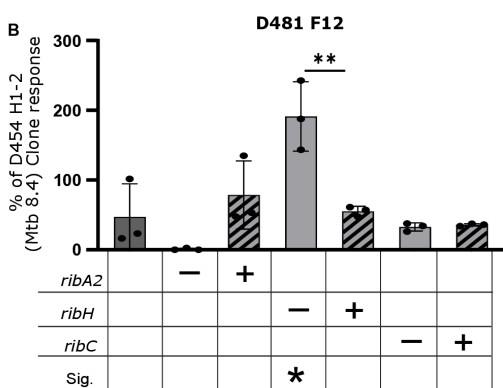

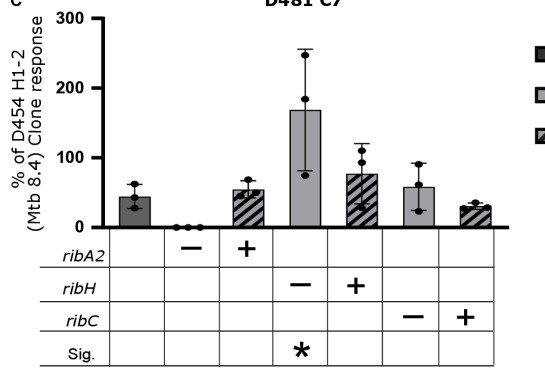

**Fig 8. Differential impacts of riboflavin pathway knockouts on MR1T recognition of Mtb.** The data represent the MR1T cell clone (1e4) IFN-γ response to dendritic cells (1e4) infected at an MOI of 10 of the indicated Mtb strain. The response for each Mtb strain was normalized to the response of D454 H1-2, a classically restricted CD8 T cell clone that recognizes Mtb8.4 [22]. Data are representative of n = 3 independent experiments. Statistical comparisons were performed using a one-way ANOVA and Sidak's multiple comparison test whereby statistical significance is represented by p < 0.05, p < 0.001, p < 0.0005 p < 0.0001, shown by *, **, ***, **** respectively.

these responses to the HLA-B45, Mtb8.4-specific clone, a classically restricted CD8 T cell clone that recognizes Mtb8.4 [27]. As observed in Msm, the deletion of *ribA2* in Mtb reduced T cell recognition of all clones by 87–97%, an effect restored by complementation (Fig 8). The residual activity seen in the Mtb *ribA2* gene knockout mutant could be blocked using an anti-MR1 antibody, suggesting the possibility that a subset of ligands are 5-A-RU independent (S13 Fig). On the other hand, the deletion of *ribH* enhanced recognition of Mtb by all three MAIT cell clones (Fig 8), which was most notable when normalized to the classically restricted CD8 T cell clone response, a different trend compared to the response observed in the single ΔribH1, ΔribH2 or double ΔribH2 ΔribH1 Msm mutants. Furthermore, knockout of the gene *ribC* had no significant effect on T cell recognition of Mtb.

## Discussion

In this study, we investigated the association between riboflavin biosynthesis and MAIT cell activation by mycobacteria using strains of Msm or Mtb in which riboflavin biosynthesis was blocked at the early (*ribA2* or *ribG*), penultimate (*ribA2* or *ribH*) or last step of the pathway (*ribC*). The dual role of *ribA2* is attributable to its bifunctionality in catalyzing the first step in the pathway via its GTP cyclohydrolase II domain, and, via its 3,4-DHBP synthase domain, the formation of a substrate of RibH, which catalyzes the penultimate step in the pathway (Fig 1). Deletion of *ribA2* or *ribG* in Msm blocks the

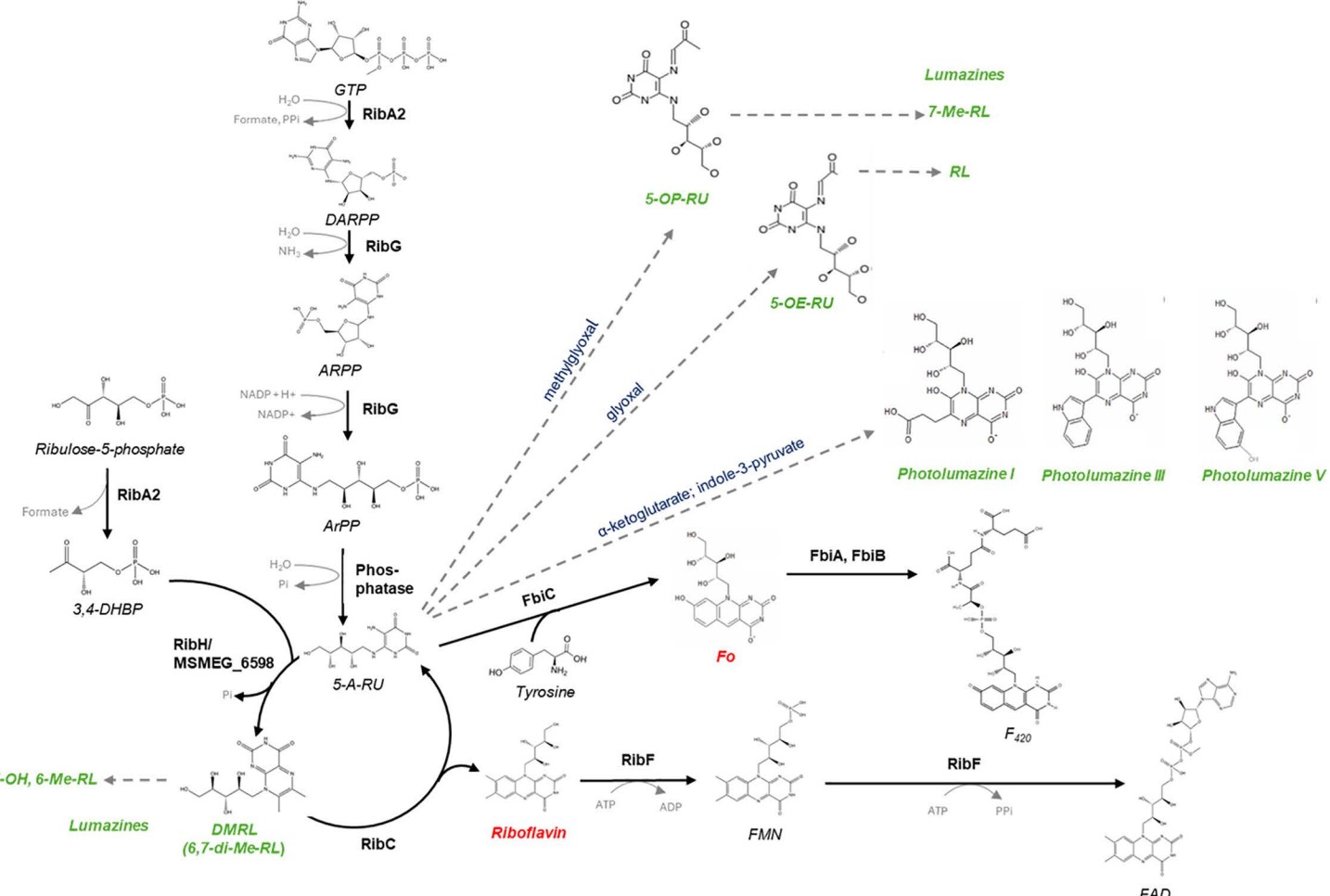

**Fig 9. MAIT cell agonists and antagonists produced via the riboflavin pathway in mycobacteria.** Solid black arrows denote enzyme-catalyzed reactions. Dashed grey arrows denote non-enzymatic reactions. MAIT cell agonists (green) or antagonists (red) are shown in bold italics. RL, ribityl-lumazine. The production of photolumazines has only been observed in Msm and not in Mtb.

production of 5-A-RU and the downstream metabolite, DMRL, whereas loss of RibA2 or RibH function by deletion in Mtb blocks the production of DMRL.

The riboflavin pathway produces both MAIT cell agonists and antagonists which differ in terms of MR1 binding affinity and potency [28]. These include intermediates ($F_0$, DMRL) and the end-product (riboflavin) of the pathway produced by enzyme-catalyzed reactions and secondary metabolites formed by non-enzymatic reactions of pathway intermediates with molecules from other pathways, which, in the case of Mtb, could include those derived from the host (Fig 9).

The level of activation of a given MR1T cell clone is therefore determined by its selectivity for MR1 ligand recognition and balance between the level of agonists *vs*. antagonists. While we have found that both $F_0$ and riboflavin are relatively weak antagonists [9], it is possible that the metabolic state of the microbe could influence the relative proportion of these ligands. The latter depends, in turn, on the metabolic state of the organism under the conditions tested. For example, the level of the toxic aldehyde, methylglyoxal, produced endogenously by Mtb [29] and also by host macrophages [30] from metabolites such as dihydroxyacetone phosphate could influence the production of 5-A-RU derived ligands. Likewise, an increased demand for $F_{420}$ for persistence of Mtb under the conditions encountered during infection (hypoxia, oxidative

stress, nitrosative stress) and/or under drug pressure [23,31] could drive the metabolic flux of 5-A-RU in mycobacteria towards the production of deazaflavins at the expense of lumazine agonists. Therefore, the agonist/ antagonist balance, and the derangement thereof by perturbing the metabolic flux through the biosynthetic pathway, could only be inferred indirectly by comparing the phenotypic readout of MR1T activation by wildtype vs. mutant strains grown under standard *in vitro* culture, albeit supplemented with riboflavin.

We previously reported the operonic organization of genes involved in this pathway in both Msm and Mtb [20]. Given this genetic linkage, it was essential to verify that disruption of individual pathway genes did not significantly impact the expression or function of other pathway members, a critical step in ensuring the rigor and accuracy of our MAIT cell activation studies. To this end, we employed a comprehensive multi-omics approach to characterize deletion mutants and their corresponding complemented strains. As anticipated, we confirmed the loss and recovery of cognate gene and protein expression in knockout mutants and complemented strains, respectively. However, we also observed unexpected effects of these genetic disruptions on the expression and abundance of other genes and proteins within the pathway. A particularly striking observation was the overexpression of the *ribH* gene in the ΔribA2 and ΔribA2::ribA2 strains of Mtb. Additionally, our multi-omics approach revealed clear distinctions between the riboflavin biosynthetic pathways in Msm and Mtb. Specifically, we noted poor expression of the *ribH* gene in complemented strains of Mtb and its homolog, *ribH1*, in Msm. Interestingly, the reduced complementation of *ribH* and *ribH1* impacted the viability of Mtb and Msm differently, with the impact on Msm being notably more severe. This result was unexpected, particularly given the presence of an additional lumazine synthase enzyme, RibH2, in Msm. Based on these findings, we hypothesized that poor complementation might result in lower availability of DMRL, subsequently affecting riboflavin biosynthesis in Msm. Indeed, our metabolic analysis demonstrated reduced riboflavin levels upon deletion of *ribH1* compared to wildtype Msm. Given the flavin-intensive metabolism of this organism, even a modest reduction in flavin availability may significantly influence essential physiological functions. Overall, our multi-omics analysis revealed that, despite observable alterations in the expression and abundance of genes and proteins within the pathway following gene deletion, the final impact on metabolic products remained minimal. As expected, deletion of genes upstream of *ribC* led to the complete loss of DMRL synthesis in both Mtb and Msm. Furthermore, the presence of a single lumazine synthase in Mtb compared to two in Msm suggests fundamental differences in physiology and relative reliance on riboflavin biosynthesis between these species. Further investigation is needed to elucidate how variations in lumazine synthase capacity influence the MR1 ligandome of these mycobacteria. Additionally, the underlying reasons for the differential contribution of *ribH1* and *ribH2* to riboflavin synthesis remain unclear and are beyond the scope of the current study.

We observed both similarities and differences between the two mycobacterial species in terms of the impact of riboflavin pathway disruption on MAIT cell activation. On the one hand – and as observed in other microbes [4] – blocking the formation of 5-A-RU by disrupting the pathway at an early stage (*ribA2*, *ribG*) had a major impact on MAIT cell activation, confirming the centrality of this pathway intermediate for production of mycobacterial MR1T agonists. While not statistically significant, we observed residual MAIT cell activation that was MR1-dependent for Msm ΔribA2 and Msm ΔribG. These data could implicate an alternate source(s) of minor mycobacterial MAIT cell agonists distinct from the riboflavin pathway. Furthermore, for both Msm and Mtb, activation of all MR1Ts tested was unaffected by knockout of *ribC* which blocks the enzymatic route of DMRL metabolism to produce riboflavin and 5-A-RU. On the other hand, deletion of *ribH* enhanced MAIT cell recognition of Mtb by all three clones, D426-G11, D481-C7 and D481-F12 whereas in Msm, the ΔribH2ΔribH1 mutant was recognized at the same level as the parental wildtype by these clones. Thus, blocking the enzymatic production of the MAIT cell agonist, DMRL, did not adversely affect MAIT cell recognition in either Mtb or Msm, and paradoxically, enhanced it in Mtb, possibly due to enhanced production of 5-A-RU-associated activating ligands. Conversely, this phenotype was not recapitulated in Msm [10].

Other studies have demonstrated variable impacts of modulating flux through the riboflavin biosynthetic pathway on MAIT cell activation. Using an alternative approach involving overexpression rather than knockout of riboflavin pathway genes, Dey

*et al.* also observed marked differences between mycobacterial species [19]. In that study, Mtb *ribH* overexpression modestly reduced activation of D426-G11, D481-C7 and D481-A9 by Mtb CDC1551 but had the opposite effect in Msm, enhancing activation of these MR1Ts up to ~ 4-fold in an MOI-dependent manner. Mtb *ribH* overexpression also enhanced by 9.3-fold the activation of D426-G11 by *M. bovis* BCG, which, unlike Mtb and Msm (and for reasons that are unclear) could not tolerate overexpression of *ribA2*, *ribG* and *ribF* [19]. In *S. typhimurium* ST313 lineage 2 strains, evasion of MAIT cell recognition was shown to be attributable to overexpression of DHBP synthase, which produces the second substrate for RibH [18].

In summary, the data presented here support a central role for the riboflavin pathway, and specifically 5-A-RU, in generating a diverse array of mycobacterial ligands, which in turn are associated with selective T cell recognition. The antigen 5-OP-RU was initially described for both *E. coli*, and *Salmonella* species, and hence supported the concept that MAIT cells were innate in their use of a highly conserved presentation molecule (MR1), a conserved TCRα chain (TRAV1–2). However, in our previous work with both Msm and Mtb, we have observed a diversity of both MR1-bound ligands and activating fractions [9]. Specifically, in our ongoing work we have not observed the lumazine antigens seen in Msm in Mtb and note that we have not observed 5-A-RU in both mycobacterial species. While failure to observe 5-A-RU could reflect their chemical instability, as observed during our assay development (S10 Fig), it does not account for the diversity of ligands that we have observed in these organisms and the ability of MR1 restricted T-cells to distinguish between bacterial species. With regards to TCR usage, we have found that the selective recognition of mycobacterial antigens is associated with TCRβ chain usage [32]. Hence, this work supports the hypothesis that further identification of these antigens could be used to define the extent to which these T cell responses are associated with infection with Mtb or could be harnessed for an improved vaccine. We also note that while ablation of RibA2 has a profound effect on the recognition of both Msm and Mtb, there were residual activating antigens in both these strains. Hence, it is likely that there remain 5-A-RU-independent antigens yet to be discovered.

## Materials and methods

### Media, culture conditions, strains, plasmids and chemicals

The plasmids used and constructs generated in this study are described in S2 Table. All strains generated were derivatives of either *E. coli* DH5α, Msm mc$^2$ 155 or Mtb H37RvMA, shown in S3 Table. *E. coli* DH5α was grown in Luria-Bertani (LB) medium (10g/L tryptone, 5g/L yeast extract and 10g/L NaCl) and plated onto Luria-Bertani agar (LB agar), containing LB media components supplemented with 15g/L agar. Mycobacterial strains were grown in Middlebrook 7H9 OADC medium (4.7 g/L 7H9 (Millipore Sigma), 2g/L glycerol (Millipore Sigma), 2.5 ml 25% Tween 80, and 100 ml of oleic acid-albumin-dextrose-catalase (OADC) enrichment (Millipore Sigma). Cells were plated onto Middlebrook 7H10 agar plates (19g/L 7H10 (Millipore Sigma), 100 ml OADC and 5 ml glycerol). Msm was grown in Erlenmeyer flasks in a rotating incubator set at 100 rpm at 37° C and unless otherwise stated Mtb was cultured in sealed cell culture flasks with no agitation at 37°C. Where required 7H9 broth or 7H10 solid media was supplemented with 50 µg/ml kanamycin, 50 µg/ml hygromycin or 2.5 µg/ml gentamicin. Riboflavin auxotrophs were supplemented with exogenous riboflavin at a concentration of 83 µM for Msm or 21 µM for Mtb based on the minimal concentration of riboflavin required to fully rescue the growth of CRISPRi hypomorphs in genes involved in riboflavin biosynthesis [20]. Media used to select for knockout mutants constructed by allelic exchange [33] or ORBIT [34] are described below.

### Construction of mutant and complemented strains

Allelic exchange by homologous recombination was used to knockout *ribC* in Mtb and *ribC*, *fbiC* and MSMEG_6598 in Msm, as described [33]. In contrast, for reasons that are unclear, attempts to generate a knockout mutant of *fbiC* in Mtb were unsuccessful. Briefly, 1000–2000 bp fragments amplified up- and downstream of the gene of interest were ligated and cloned in p2NIL. The *Pac*I cassette from either pGOAL19 or pGOAL17 cloned in the *Pac*I site in the p2NIL (S2 Table). The resulting suicide construct was electroporated into the mycobacterial strain and single crossover clones selected by

plating onto 7H10 media supplemented with 50 µg/ml kanamycin, 50 µg/ml hygromycin and 40 µg/ml X-gal (for pGOAL19 constructs) or 50 µg/ml kanamycin and 40 µg/ml X-gal (for pGOAL17 constructs). Double crossover recombinants were selected by plating onto media supplemented with 2% sucrose, 40 µg/ml X-gal and riboflavin at concentration of 83 µM for Msm or 21 µM for Mtb. The ORBIT system [34] was used to create individual gene knockouts of *ribA2*, *ribG* or *ribH* in Msm and *ribA2* or *ribH* in Mtb although an attempt to construct a *ribG* knockout in Mtb using ORBIT was unsuccessful. Briefly, a targeting oligonucleotide containing 70 bp of homology to the target gene flanked by the BxbI phage *attP* sequence (S4 Table) was co-electroporated with pKM464 into the mycobacterial strain carrying pKM461 and selected on 7H10 agar containing 10% sucrose and 50 µg/ml hygromycin. To create complementation vectors, ca. 300–400 bp upstream of the gene along with its open reading frame were amplified and cloned into an integrating vector. All primers and amplicons are described in S4 Table. The complementation vectors were introduced into the knockout mutants by electroporation and selected on the appropriate antibiotic selection plates. DNA was extracted from all gene knockout and complement strains using the CTAB method [35] and the gene deletion/ gene restoration verified by whole genome sequencing.

### RNA extraction

Mtb and Msm strains were inoculated in 80 ml of Middlebrook 7H9 OADC broth supplemented with 0.01% tyloxapol until mid-log phase ($OD_{600}$ between 0.4 and 0.6) was reached. Media was supplemented with riboflavin as necessary. Afterwards, cells were pelleted, resuspended in 1 ml of TriZOL (Thermo Scientific) and 5 µl of glycogen (Thermo Scientific) and then transferred to screw-cap homogenization microtubes prefilled with Lysing Matrix B (MP Biomedical). The cells were homogenized using the FastPrep-24 bead beater (MP Biomedical) at 6 m/s for 30 seconds (6 rounds), cooling on ice for 2 min between cycles. After centrifugation at 12,000 × *g* for 120 seconds, homogenate was collected in a new tube and treated with 200 µl of chloroform followed by vortexing for 15 seconds and incubation at room temperature for 5 minutes. Samples were centrifuged at 12,000 × *g* for 15 minutes at 4 °C and upper aqueous phase was transferred into new tubes containing 500 µl of cold isopropanol for overnight precipitation. Samples were centrifuged at 12,000 × *g* for 10 min and supernatant disposed. Pellet was washed with 1 ml of cold 80% ethanol and then resuspended with nuclease free water. Extracted RNA was processed for DNase treatment and clean-up using the Rneasy Mini Kit (QIAGEN) and Rnase free Dnase Set (QIAGEN) according to manufacturer's instruction to eliminate genomic DNA. RNA was quantified using a NanoDrop (Thermo Scientific) and Agilent TapeStation.

### Gene expression analysis

Two hundred ng of RNA was retrotranscribed to complementary DNA (cDNA) using the SuperScript IV VILO Master Mix (Thermo Scientific) according to manufacturer's instruction. PowerTrack SYBR green Master Mix (Thermo Scientific) with 250 nM of primer pair was used to amplify the target region of interest for 5 ng of cDNA following manufacturer's instruction. Quantification was run on the QuantStudio 3 Real-Time PCR System (Thermo Scientific). A no reverse transcriptase (no-RT) control was included for all samples to ensure that there was no genomic DNA contamination. Amplicon size was confirmed by melting curves for each sample. All qRT-PCR experiments were performed with two technical replicates derived from three biological replicates for Mtb and two biological replicates for Msm. Transcript levels of the target genes were normalized to the housekeeping gene *sigA* (an essential housekeeping gene which is stably expressed) and calibrated to wildtype Mtb and Msm to calculate the ΔΔCt and determine fold difference in gene expression. The primers used are listed in S5 Table.

### Growth kinetics of mycobacteria

Msm and Mtb growth kinetics in 7H9 media were monitored by $OD_{600}$ as previously described [20].

## Growth and preparation of Mtb and Msm for targeted proteomics

Glycerol stocks of Mtb and Msm strains were used to start cultures on 7H11 agar plates (Millipore Sigma, Cat No.: M0428) supplemented with the appropriate antibiotics with or without riboflavin. Plates were incubated for 2 weeks for Mtb and 4–5 days for Msm at 37 °C. Cells were scraped from plates into 7H9 media supplemented with OADC with or without riboflavin and grown at 37°C until mid-log phase. Mtb cells from all strains were inactivated by γ-irradiation and inactivation was confirmed using the Alamar Blue assay. Cells were harvested and washed 3 times either with phosphate buffered saline (PBS) or water to remove media and stored at -80 °C until further processing. Cell pellets were further processed as previously described [20].

## Growth and preparation of Mtb and Msm for targeted metabolomics and untargeted proteomics

Glycerol stocks of Mtb and Msm strains were used to start cultures on 7H11 agar plates supplemented with the appropriate antibiotics with or without riboflavin. Plates were incubated for 2 weeks for Mtb and 4–5 days Msm at 37 °C. Cells were scraped from plates into 7H9 media supplemented with OADC with or without riboflavin and grown at 37°C until mid-log phase. Cells were collected via centrifugation, washed with mass spectrometry grade water and resuspended in Acetonitrile/Methanol/Water (2:2:1) and then transferred to screw-cap homogenization microtubes prefilled with 0.1 mm zircon beads. The cells were homogenized using the FastPrep-24 bead beater (MP Biomedical, SKU:116004500) at 6 m/s for 30 seconds (10 rounds), cooling on ice for 2 minutes between cycles. Homogenized cells were collected and spun down at 16,000 × $g$ for 20 minutes and supernatant collected for metabolomics and pelleted cell lysate for proteomics. The pelleted proteins were quantified using Bicinchoninic acid assay (Thermo Scientific, Cat No.: 23225). Supernatant and pellet were then stored at -20°C until further processed.

## Liquid chromatography - MS/MS analysis (Proteomics)

Thirty µg of retentate from whole cell lysate from Msm and Mtb after ultrafiltration with 3 kDa filters (Millipore Sigma, Product Number: UFC9003), was subjected to in-gel trypsin digestion as described previously [36]. Digested samples were resuspended in 120 µl of loading buffer (3% acetonitrile, 0.1% formic acid in water). All samples were analyzed using nano-UHPLC (ultrahigh-pressure liquid chromatography) nanoElute (Bruker Daltonics, USA) coupled to a timsTOF Pro (Bruker Daltonics, USA) mass spectrometer. 0.25 µg of total peptides were loaded onto a trap cartridge using PepMap Neo Trap Cartridge (Thermo Scientific, Cat: 174500) (5 µm C18 300 µm X 5 mm) packed with spherical, fully porous, ultrapure silica and reverse phase UHPLC was performed in a PepSep Column (Bruker Daltonics, Part No:1895802) (1.5 µm C18 75 µm X 25 cm). Mobile phase A was 0.1% formic acid in water; mobile phase B was 0.1% formic acid in acetonitrile. Liquid chromatography (LC) separation was carried out at a flow rate of 0.5 µL/min using a linear gradient of 5% solvent B to 30% solvent B over a duration 17.8 minutes, then 30% solvent B to 95% solvent B for 2.5 minutes. Thereafter, 95% solvent B was held for 2.4 minutes for equilibration.

Mass spectra and tandem mass spectra were obtained using the parallel accumulation serial fragmentation (PASEF) acquisition method in positive ion mode. Captive spray source was operated with the following settings: 1600 V capillary voltage, 3.0 l/min of dry gas at 200°C with the NanoBooster switched off.

A spectral library was generated by analyzing samples using data-dependent acquisition mode (DDA-PASEF). DDA was set to obtain high resolution MS scans over the m/z range from 100 to 1700 and ion mobility range from 0.70 Vs/cm$^2$ to 1.50 Vs/cm$^2$. Ramp time was set at 100 msec and accumulation time was set at 100 msec. The collision energy was set to follow a linear function starting from 20 eV for 0.6 V. s/cm$^2$ to 59 eV for 1.6 V. s/cm$^2$. The number of PASEF MS/MS was set to 6. For DIA-PASEF analysis, data was acquired with a method consisting of 24 mass steps per cycle (S6 Table). Mass width was set at 50 Da with 1 Da overlap between windows covering a mass range between 307.8 to 1484.8 Da. The ion mobility range (1/K$_0$) covered was between 0.71 to 1.44. The collision energy was set to follow a linear function starting from 20 eV for 0.6 V. s/cm$^2$ to 59 eV for 1.6 V. s/cm$^2$.

## Data analysis of untargeted proteomics Data Independent Acquisition (DIA)

DIA-PASEF data was processed on FragPipe software v. 19.1 (MSFragger version 3.7, IonQuant version 1.8.10, Philosopher version 4.8.1) using the DIA_SpecLib-Quant workflow as described [37]. DIA raw files were loaded with DIA Quant selected as data type while DDA raw files were loaded for spectral library generation. A search was conducted using default settings with the FASTA sequence obtained from the Mycobrowser database (Release 4, 2021-03-23). Decoy and contaminant sequences were generated by FragPipe. Default settings of MSFragger was used, including oxidation of M (+ 15.9949 Da mass shift) acetylation at N-terminal (+ 42.0106) as variable modifications. Advanced output options were set to write calibrated mzML and MGF files. Tryptic searches were performed with 20 ppm precursor and fragment tolerance permitting up to 2 missed cleavages. Search results were filtered to 1% FDR at both protein and peptide levels using peptide prophet. Output files with quantification based on summed intensity of all ions for a single peptide and summed intensity of peptides for a single protein were generated. The output file containing summed intensity of peptides for a single protein was used for quantification of proteins and is provided in S6 Table.

## Targeted proteomics analysis by parallel reaction monitoring -PASEF Mass Spectrometry (PRM-PASEF MS)

DDA peptide search results from MSFragger version 3.7 were used to build a spectral library in Skyline [38]. The Ion Mobility library was generated on Skyline software from raw DDA files. The protein list was trimmed to contain only proteins from the riboflavin biosynthetic pathway. Tryptic peptides were filtered to have length only be between 8 and 30 amino acids. Tryptic peptides containing C, M, D-P, and W were excluded. Peptides for each protein were further screened based on the shape of the precursor chromatogram and absence of the precursor peak from the cognate knock-out sample. Ion types were limited to only precursor ions (2 and 3 ion charges) and 6 y ions (1 and 2 ion charges). The retention time window was set at 3 minutes and mass accuracy set at 10 ppm for both MS1 and MS2 filtering. The method was exported to timsTOF engine (Bruker Daltonics, USA) as an unscheduled method. 0.25 µg of each Mtb sample was loaded for reverse phase UHPLC and MS. performed. Analytical settings were kept the same as previously described (LC-MS/MS Section) with the inclusion of a target list containing the Mtb peptides to quantify in the PRM-PASEF method generated.

Raw data were exported into Skyline software and peptides with the best peak shape and intensity were selected for further optimization. At least, one peptide per protein was selected for the final method. For absolute quantification, peptides labeled with heavy C-terminal Lysine (K) or Arginine (R) were purchased (Biosynth Ltd, UK) to be used as internal standards. Limit of detection (LOD) and Limit of quantification (LOQ) were calculated using a linear regression method as described elsewhere [39]. Details of the peptide sequences and transition list used for the PRM-PASEF experiment are shown in S7 Table.

## Targeted quantification of riboflavin pathway proteins by PRM-PASEF MS

A mix of the heavy labeled peptides containing $10^{-1} - 10^{-5}$ pmol/µl of each peptide was used to resuspend each of the digested samples at a final concentration of 250 ng/µl of digested protein (S8 Table). After resuspension, the sample peptide concentration was estimated using a Nanodrop 1000c and the concentration was adjusted to ensure equal loading of all samples. The developed PRM-PASEF MS method was exported as a scheduled method and used to obtain the Ratio to Heavy Standard for each of the monitored peptides. Briefly, raw data from each injection was imported into Skyline [38] and peak boundaries for each peptide were manually validated and adjusted if necessary. The Ratio to Heavy Standard for each peptide as calculated by Skyline was exported into Prism GraphPad for statistical analysis.

## Liquid chromatography -MS/MS analysis (Metabolomics)

Supernatant collected for metabolomics was dried in savant and their weight determined. Dried samples were then resuspended in 3% Acetonitrile in Water (0.1% Formic Acid) to make a final concentration of 2 mg/ml for Mtb and 10 mg/ml for Msm. Samples were analyzed using the Agilent 1290 Infinity III Liquid Chromatography System

attached to a 6470 Triple Quadrupole MS system. 10 µl of samples were loaded per injection. Reverse phase UHPLC was performed using InfinityLab Poroshell 120 EC-C18 Column (Agilent, Part Number: 693975–906) (4.6 x 150 mm, 2.7 µm) for Mtb samples and InfinityLab Poroshell 120 EC-C18 Column (Agilent, Part Number: 695975–906) (4.6 × 100 mm, 2.7 µm) for Msm samples. Mobile phase A was 0.1% formic acid in water; mobile phase B was 0.1% formic acid in acetonitrile. LC separation was carried out at a flow rate of 0.3 µL/min using a linear gradient of 2% solvent B held for 1.5 minutes and then to 20% solvent B over a duration 5 minutes, then 20% solvent B to 99% solvent B for 3 minutes. 99% solvent B was held for 0.5 minutes and then returned to 2% solvent B for 3 minutes for equilibration.

Mass spectra and tandem mass spectra were carried out using the dynamic multiple reaction monitoring (DMRM) acquisition method in positive ion mode. Column temperature was set at 45 ºC. Source was operated with the following settings: 10 l/min of gas flow at 325ºC. Nebulizer was set at 20 psi. Sheath gas temperature was set at 350 ºC with flow rate of 10 l/min. Capillary voltage was set at 3500 V and nozzle voltage set at 0 V.

### Targeted metabolomics analysis by dynamic multiple reaction monitoring-mass spectrometry (DMRM-MS)

Agilent Masshunter Workstation was used to build MRM and DMRM methods for all compounds. Standards for DMRL, 5-A-RU, and riboflavin at 1 µg/ml were used to determine optimal fragmentor voltage and collision energy for each compound. Subsequently, observed product ion were filtered to keep the top three ions based on abundance. An unscheduled MRM was run to determine retention time for each compound. Final retention times were set in DMRM mode, and a retention time window was set at 2 minutes. See S9 Table for parent ion and product ion list used for DMRM. Mixtures containing 103 ng/ml to 10–3 ng/ml of the four compounds were analyzed using the DMRM method to determine the limit of detection (LOD) and limit of quantification (LOQ) as reported in S9 Table. The product ion for each parent ion with the highest abundance was selected as the quantitative ion while the two other product ions in addition to the most abundant product ion were used to qualitative ions (S9 Table).

Samples for Mtb and Msm were loaded as described for LC-MS/MS analysis. Raw data for each injection was imported into Agilent Masshunter Qualitative Analysis Software version 10.0. Peaks for the three selected product ions were integrated for each compound and adjusted if necessary. Total Area Under Curve (AUC) was manually extracted for each product ion. To ensure accurate quantification, detected compounds were qualified by comparing the ratio of the peak area of the transition ions observed in the sample to the ratio observed for the standards. Ratios were compared as dot products and a cutoff of 0.98 was set for qualification. Extracted data that passed the qualification step was analyzed on GraphPad Prism. MRM data have been deposited at the Dryad Digital Repository [40] and are available at: https://doi.org/10.5061/dryad.z08kprrr1.

### Whole genome sequencing (WGS)

Genomic DNA was analyzed at the Genomics and Microarray Core at the University of Colorado Anschutz using a NovaSEQ 6000 Instrument, paired end 150 cycles 2x150. Reads were subjected to quality control using FastQC before and after trimming. Trimming was performed using trimmomatic. Alignment of reads to the Mtb H37Rv genome (NC_000962.3) or Msm mc$^2$ 155 (CP000480.1)/ (NZ_CP054795.1) was performed using Bowtie2. Due to the high density of reads, the aligned sam files were subjected to subsample using samtools. Samtools were also used to determine coverage results. Subsample alignment was used to determine SNPs using lofreq with a minimum coverage of 10 and minimum frequency of 80. The bam file was visualized using the Geneious software. The expected deleted regions were confirmed by visual inspection and very few inadvertent second-site mutations were detected in the deletion mutants of Msm and Mtb relative to their parental wildtype strains (S10, S11 and S12 Tables). WGS data have been deposited at the at the Dryad Digital Repository [41,42] and are available at: https://doi.org/10.5061/dryad.4tmpg4fnv and https://doi.org/10.5061/dryad.vq83bk454

## T cell clones

All T cell clones were expanded and maintained as previously described [7,27]. Briefly, a T cell clone is incubated with irradiated PBMC and LCL in addition to anti-CD3 for 12–17 days and frozen. The anti-CD3 is washed out on day 5. Four T cell clones were used in this work. All have been published previously: D426 G11, D481-C7, and D481-F12 are MAIT cell clones expressing TRAV1–2 [7,9], and D454 H1-2 is a CD8 T cell clone that recognizes Mtb8.4$_{33-43}$ and is restricted by HLA-B*15:01 [27] (see S1 Table for details of TCRs). D426 and D454 were isolated from a patient that was Mtb-infected (PPD+) and D481 from a patient that had previously had active TB. To generate these clones, CD8+T cells were sorted from PBMC and incubated in a 96 well limited dilution assay with autologous Mtb-infected dendritic cells for 2 weeks. Wells with growth were screened against autologous and heterologous Mtb-infected dendritic cells. T cell clones that responded only to autologous dendritic cells were characterized to determine the Mtb antigen they recognize (D454 H1-2), and T cell clones that responded to both autologous and heterologous dendritic cells were blocked with various antibodies. D426 G11, D481 C7, and D481 F12 were blocked by anti-MR1 antibody and later found to stain the MR1/5-OP-RU tetramer.

## IFN-γ ELISpot assays

All Msm and Mtb strains were grown in the presence of riboflavin with or without supplements as needed (see above) to an OD$_{600}$ ~0.5 and frozen down with 10% glycerol. All strains were titered on 7H10 agar plates to obtain CFU/ml. Monocyte derived dendritic cells were prepared from D520 PBMC, which expresses HLA-B*15:01, using plastic adherence and culturing for 5 days with IL-4 and GM-CSF [27]. For all ELISPOT assays, ELISPOT plates were coated with anti-IFN-γ antibody (Mabtech, clone 1-D1K, Cat No. 3420-3-1000, used at 10 µg/ml) as previously described [43]. After overnight incubation at 4°C, ELISPOT plates were washed three times with phosphate-buffered saline and then blocked with RPMI 1640+10% human serum for at least 1 hour. For experiments with Msm approximately 1×10$^4$ dendritic cells were plated in the ELISPOT wells in RPMI+10% HuS. Msm bacilli were added to the wells at concentrations indicated for each experiment. PHA (Millipore) was used as positive control. After at least 1 hour, 1–2×10$^4$ T cell clones that were freshly thawed from a cryopreserved expansion were added, and the plate was incubated overnight at 37°C. IFN-γ ELISPOTs were enumerated following development as previously described [43]. For experiments with live Mtb, 5×10$^5$ dendritic cells were plated in each well of an ultralow adherence 24 well tissue culture plate. A vial of frozen Mtb was thawed, passaged through a tuberculin needle >20 times, and the appropriate volume was added to the dendritic cells at the MOI indicated and incubated overnight. Dendritic cells were harvested and 1×10$^4$ uninfected or Mtb-infected dendritic cells were plated in the ELISPOT wells in RPMI+10% HuS. 1–2×10$^4$ T cell clones were added, and the plate was incubated overnight at 37° C. IFN-γ ELISPOTs were enumerated following development.

## Statistical analyses

Statistical calculations were performed using GraphPad Prism 10 statistical software (GraphPad Software, Inc., San Diego, CA). For mycobacterial growth on solid media, statistical comparisons against the corresponding wildtype strain were performed using a one-way ANOVA and Kruskal-Wallis comparison test. For growth in liquid culture, comparisons were performed using area under the curve (for Msm) or one-way ANOVA and either Sidak's or Dunnett's multiple comparison test (for Mtb). For proteomics, qRT-PCR and the EliSPOT assay, statistical comparisons were performed using a one-way ANOVA followed by Dunnett's or Sidak's multiple comparison test. The latter analyses were done using GraphPad Prism 9 and statistical significance was established at $p < 0.05$ or $p < 0.001$.

## Supporting information

**S1 Fig. Schematic of genomic loci of the riboflavin biosynthesis pathway knockout mutants of Msm (A) and Mtb (B) constructed by ORBIT or homologous recombination.** The mutants constructed by ORBIT carry a vector sequence which included a *hyg* resistance marker whereas the allelic exchange mutants are unmarked.
(TIF)

PLOS Pathogens

**S2 Fig. Growth of wildtype, knockout and complemented mutant strains of Msm on 7H10 media.** Msm cultures were diluted 10-fold, spread onto 6-well plates and incubated for 4 days (**A**) and CFUs enumerated. (**B**). Msm Δ*ribA2*::*ribA2*, Δ*ribH1*::*ribH1*, Δ*ribH2*Δ*ribH1*::*ribH2* and Δ*ribH2*Δ*ribH1*::*ribH2 ribH1* plates were incubated for an additional day as colonies were too small to count on day 4. The colony sizes of all emerging colonies were measured (**C**). Riboflavin supplement was used at a concentration of 83 µM for culturing Msm. Error bars represent standard deviation from two biological replicates. Statistical comparisons against wild-type were performed using a one-way ANOVA and Kruskal-Wallis comparison test whereby statistical significance is represented by $p < 0.05$, shown by an asterisk.
(TIF)

**S3 Fig. Phenotypic growth of wildtype, knockout and complemented mutant strains of Mtb on 7H10 media.** Mtb cultures were diluted 10-fold, spread onto 6-well plates and incubated for 4 weeks (**A**) and CFUs enumerated (**B**). The colony sizes of all emerging colonies were measured (**C**). Riboflavin supplement was used at a concentration of 21 µM for Mtb. Error bars represent standard deviation from two biological replicates. Statistical comparisons, against wild-type were performed using a one-way ANOVA and Kruskal-Wallis comparison test whereby statistical significance is represented by $p < 0.05$, shown by an asterisk.
(TIF)

**S4 Fig. Quantification of protein levels of RibA2, RibG, RibH1, RibH2, RibC and FbiC in Msm strains Δ*fbiC*, Δ*fbiC*::*fbiC* and Δ*ribH2*Δ*ribH1*::*ribH2*.** Protein abundance was measured using DIA proteomics (n = 3 biological replicates, 2 technical replicates). Data are shown as mean quantification ± SEM. Statistical comparisons were performed using a one-way ANOVA and Sidak's multiple comparison test whereby statistical significance is represented by $p < 0.05$, $p < 0.001$, $p < 0.0005$ $p < 0.0001$, shown by *, **, ***, **** respectively. Only statistically significant relationships are shown. RF, riboflavin. Sig., significance in comparison to wild type (+/- RF) is shown as symbols as described in legend ($p < 0.05$). Wild type ± RF shown in this figure are the same as those in Figs 3 and 4.
(TIF)

**S5 Fig. Quantification of *fbiC* transcript and FbiC protein levels in Msm strains. (A)** Protein abundance was measured using DIA proteomics (n = 2 biological replicates, 2 technical replicates). Data are shown as mean quantification ± SEM. **(B)** Fold changes in *sigA*-normalized transcript levels relative to wildtype. Transcript levels of the target genes were normalized to the housekeeping gene *sigA* (an essential housekeeping gene which is stably expressed) and scaled to the average of wildtype Msm to calculate the ΔΔCt and determine the fold difference in gene expression (n = 3 biological replicates, 2 technical replicates). Statistical comparisons were performed using a one-way ANOVA and Sidak's multiple comparison test whereby statistical significance is represented by $p < 0.05$, $p < 0.001$, $p < 0.0005$ $p < 0.0001$, shown by *, **, ***, **** respectively. Only statistically significant relationships are shown. RF, riboflavin. Sig., significance in comparison to wild type (+/- RF) is shown as symbols as described in legend ($p < 0.05$).
(TIF)

**S6 Fig. Protein abundance of Mtb strains assessed by PRM.** Stable isotope-labeled standard (SIS) peptides were selected for RibC and RibH for targeted quantification of proteins using LC-MS/MS. Ratio to heavy standard represents protein abundance of RibC (**A**), and RibH (**B** and **C**) in the different strains of Mtb. Data are shown as mean quantification ± SEM of three biological replicates. Statistical comparisons were performed using a one-way ANOVA and Sidak's multiple comparison test whereby statistical significance is represented by $p < 0.05$, $p < 0.001$, $p < 0.0005$ $p < 0.0001$, shown by *, **, ***, **** respectively. Only statistically significant relationships are shown. RF, riboflavin. Sig., significance in comparison to wild type (+/- RF) is shown as symbols as described in legend ($p < 0.05$).
(TIF)

**S7 Fig. Detection of DMRL and RF in Msm strains.** Representative chromatograms depicting the most intense multiple reaction monitoring (MRM) transitions for DMRL (A, m/z 327.1→193) and RF (B, m/z 377.2→243.1) in Msm strains. Each metabolite was monitored using its most intense transition ion, along with the next two most intense transition ions, to confirm specificity.
(TIF)

**S8 Fig. Quantification of riboflavin in Msm and Mtb strains.** Levels of intracellular riboflavin were quantified in Msm (**A** and **B**) and Mtb (**C**) using MRM. Data are shown as mean quantification ± SEM for three biological replicates. Quantification was carried out using chromatographic peak area of the most intense transition ion (m/z 377.2→243.1) of RF. Statistical comparisons were performed using a one-way ANOVA and Sidak's multiple comparison test whereby statistical significance is represented by $p < 0.05$, $p < 0.001$, $p < 0.0005$ $p < 0.0001$, shown by *, **, ***, **** respectively. Only statistically significant relationships are shown. RF, riboflavin. Sig., significance in comparison to wild type (+/- RF) is shown as symbols as described in legend ($p < 0.05$). Statistical comparison was done only between samples not spiked with exogenous riboflavin.
(TIF)

**S9 Fig. Detection of DMRL and riboflavin in Mtb strains.** Representative chromatograms depicting the most intense multiple reaction monitoring (MRM) transitions for DMRL (**A**, m/z 327.1→193) and riboflavin (RF) (**B**, m/z 377.2→243.1) in Mtb strains. Each metabolite was monitored using its most intense transition ion, along with the next two most intense transition ions, to confirm specificity.
(TIF)

**S10 Fig. Quantification of Standards of 5-A-RU, riboflavin and DMRL.** Stability of 5-A-RU (**A**, **B**, and **C**), riboflavin (**D**, **E**, and **F**) and DMRL (**G**, **H**, and **I**) were monitored using the three most intense transition ions over three injections with samples kept at 5°C with ≈ 11 minutes between injections. Data are shown as percentage of chromatographic peak area of initial injection.
(TIF)

**S11 Fig. Impact of riboflavin pathway mutations on MR1T recognition of Msm is MR1 dependent.** MR1T cell clone (1e4) IFN-γ response to dendritic cells (1e4) incubated with Msm wildtype, Δ*ribA2*, or Δ*ribG* at a MOI of 5 or MOI of 1 is inhibited by anti-MR1 antibody blockade (aMR1 or isotype, 2µg/ml). Response was normalized to wildtype Msm no antibody condition. Data are representative of n = 3 independent experiments. (**A**) D426 G11 at MOI 5, (**B**) D426 G11 at MOI 1, (**C**) D481 F12 at MOI 5, (**D**) D481 F12 at MOI 1, (**E**) D481 C7 at MOI 5, (**F**) D481 C7 at MOI 1. Statistical comparisons were performed using a one-way ANOVA and Sidak's multiple comparison test whereby statistical significance is represented by $p < 0.05$, $p < 0.001$, $p < 0.0005$ $p < 0.0001$, shown by *, **, ***, **** respectively.
(TIF)

**S12 Fig. Wildtype Msm and PHA control are concordant for all experiments conducted for Fig 7. The riboflavin pathway mutants were normalized to the wildtype Msm.** Shown are the spot counts from MR1T cell clones (1e4) IFN-γ response to DCs (1e4) for the wildtype Msm (MOI 10) and PHA (10ug/ml) conditions for every experiment included in Fig 7. Statistical comparisons were performed using a one-way ANOVA and Sidak's multiple comparison test whereby statistical significance is represented by $p < 0.05$, $p < 0.001$, $p < 0.0005$ $p < 0.0001$, shown by *, **, ***, **** respectively.
(TIF)

**S13 Fig. Impact of riboflavin pathway mutations on MR1T recognition of Mtb is MR1 dependent.** MR1T cell clone (1e4) IFN-γ response to dendritic cells (1e4) incubated with Mtb wildtype, Δ*ribA2* or Δ*ribH* at a MOI of 5 is inhibited by anti-MR1 antibody blockade (aMR1 or isotype, 2µg/ml). Response was normalized to wildtype Mtb no antibody condition.

Data are representative of n = 3 independent experiments. Statistical comparisons were performed using a one-way ANOVA and Sidak's multiple comparison test whereby statistical significance is represented by $p < 0.05$, $p < 0.001$, $p < 0.0005$ $p < 0.0001$, shown by *, **, ***, **** respectively.
(TIF)

**S1 Table. TCR sequence of MR1T cell clones.**
(DOCX)

**S2 Table. Plasmids used in this study.**
(DOCX)

**S3 Table. Strains used in this study.**
(DOCX)

**S4 Table. Oligonucleotides used to create knockouts and complemented strains.**
(DOCX)

**S5 Table. Primers used for qRT-PCR analysis.**
(DOCX)

**S6 Table. Window scheme used for DIA-PASEF acquisition.**
(DOCX)

**S7 Table. Peptides used for targeted proteomics by PRM-MS.**
(DOCX)

**S8 Table. Peptides used for targeted proteomics by PRM-MS and concentration used in final assay.**
(DOCX)

**S9 Table. Precursors and transition ions used for targeted metabolomics by MRM-MS.**
(DOCX)

**S10 Table. Confirmation of riboflavin pathway mutants by WGS.**
(DOCX)

**S11 Table. SNPs detected in Msm knockout mutants and not in wildtype.**
(DOCX)

**S12 Table. SNPs detected in Mtb mutants and not in wildtype.**
(DOCX)

**S1 Data. All data used for statistical analyses and generation of main figure graphs.**
(XLSX)

**S2 Data. All data used for statistical analyses and generation of supplementary figure graphs.**
(XLSX)

**S3 Data. All data used for statistical analyses and generation of MR1T recognition of Msm (Fig 7).**
(XLSX)

**S4 Data. All data used for statistical analyses and generation of MR1T recognition of Mtb (Fig 8).**
(XLSX)

## Author contributions

**Conceptualization:** Melissa D. Chengalroyen, Digby F. Warner, Deborah A. Lewinsohn, Karen M. Dobos, Valerie Mizrahi, David M. Lewinsohn.

**Data curation:** Melissa D. Chengalroyen, Nurudeen Oketade, Gwendolyn M. Swarbrick.

**Formal analysis:** Melissa D. Chengalroyen, Nurudeen Oketade, Gwendolyn M. Swarbrick, Carolina Mehaffy.

**Funding acquisition:** Deborah A. Lewinsohn, Erin J. Adams, William Hildebrand, Karen M. Dobos, Valerie Mizrahi, David M. Lewinsohn.

**Investigation:** Melissa D. Chengalroyen, Nurudeen Oketade, Aneta Worley, Megan Lucas, Mabule L. Raphela, Gwendolyn M. Swarbrick, Mandisa Zuma, Carolina Mehaffy.

**Methodology:** Melissa D. Chengalroyen, Nurudeen Oketade, Luisa Maria Nieto Ramirez, Gwendolyn M. Swarbrick, Paul S. Soma, Carolina Mehaffy, Karen M. Dobos, Valerie Mizrahi, David M. Lewinsohn.

**Project administration:** Melissa D. Chengalroyen, Nurudeen Oketade, Gwendolyn M. Swarbrick, Carolina Mehaffy, Karen M. Dobos, Valerie Mizrahi, David M. Lewinsohn.

**Resources:** Karen M. Dobos, Valerie Mizrahi, David M. Lewinsohn.

**Supervision:** Melissa D. Chengalroyen, Luisa Maria Nieto Ramirez, Paul S. Soma, Digby F. Warner, Deborah A. Lewinsohn, Karen M. Dobos, Valerie Mizrahi, David M. Lewinsohn.

**Validation:** Melissa D. Chengalroyen, Gwendolyn M. Swarbrick.

**Visualization:** Melissa D. Chengalroyen, Nurudeen Oketade, Gwendolyn M. Swarbrick, Carolina Mehaffy, Valerie Mizrahi.

**Writing – original draft:** Melissa D. Chengalroyen, Nurudeen Oketade, Megan Lucas, Luisa Maria Nieto Ramirez, Valerie Mizrahi, David M. Lewinsohn.

**Writing – review & editing:** Melissa D. Chengalroyen, Nurudeen Oketade, Karen M. Dobos, Valerie Mizrahi, David M. Lewinsohn.

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
