## [Decision Letter · Decision Letter 0]

27 Sep 2024

Response to Reviewers
Revised Manuscript with Track Changes
Manuscript

Editor-in-Chief

PLOS Pathogens

**Journal Requirements:**

1) We do not publish any copyright or trademark symbols that usually accompany proprietary names, eg ©,  ®, or TM  (e.g. next to drug or reagent names). Therefore please remove all instances of trademark/copyright symbols throughout the text, including:

- TM on page: 20.

2) We note that your Data Availability Statement is currently as follows: "All relevant data are within the manuscript and its Supporting Information files". Please confirm at this time whether or not your submission contains all raw data required to replicate the results of your study. Authors must share the “minimal data set” for their submission. PLOS defines the minimal data set to consist of the data required to replicate all study findings reported in the article, as well as related metadata and methods (https://journals.plos.org/plosone/s/data-availability#loc-minimal-data-set-definition).

3) Please ensure that the funders and grant numbers match between the Financial Disclosure field and the Funding Information tab in your submission form. Note that the funders must be provided in the same order in both places as well. Currently, these funds "Oppenheimer Fellowship from the Oppenheimer Memorial Trust, the Broad Institute, and internal funds from Colorado State University" are missing from the Funding Information tab. 

Please indicate by return email the full and correct funding information for your study and confirm the order in which funding contributions should appear. Please be sure to indicate whether the funders played any role in the study design, data collection and analysis, decision to publish, or preparation of the manuscript.

**Reviewers' Comments:**

**Part I - Summary**

Reviewer #1: Chengalroyen and Oketade et al generate knock-out mutants of genes involved in riboflavin biosynthesis in Mycobacterium smegmatis and M. tuberculosis and reconstitute knock-outs with the relevant genes. They evaluate the impact of the mutants on the growth of the bacteria in the presence and absence of riboflavin, quantify the impact of individual knock-outs on riboflavin biosynthesis gene transcripts and protein levels and test their capacity to stimulate previously generated M. smegmatis and M. tuberculosis-specific MR1-reactive T cell clones. They confirm a central role for 5-A-RU in M. smegmatis and M. tuberculosis-derived MAIT cell antigens. They also argue that residual activation by mutants of RibA2 or RibG, required for the generation of 5-A-RU, suggests an alternate source/s of minor mycobacterial MAIT cell agonists. They found that deletion of RibH which catalyses the conversion of 5-A-RU and 3,4-DHBP to DMRL enhanced MAIT cell recognition of M. tuberculosis but had no impact on the MAIT cell response to M. smegmatis.

Altogether this study adds to similar studies pinpointing 5-A-RU as the key intermediate in riboflavin biosynthesis antigen generation in a range of bacteria. Mycobacteria is of clinical relevance and the study is hence important.

My main concerns relate to some of the mutants generated where reconstitution was little or not successful, shedding doubt on the specificity of the mutants and hence their interpretations of T cell activation. For the T cell activation experiments raw activation data needs to be shown, stats to wild-type bacteria activation done and MR1 dependency tested. These experiments were sometimes over-interpreted, especially for the TRAV1-2 neg T cell clone where the error bars are too large for some mutants to interpret the data at all.

Reviewer #2: The authors of the study conducted experiments on mutants of Mycobacterium smegmatis (Msm) and Mycobacterium tuberculosis (Mtb) to explore the role of genes related to the riboflavin biosynthesis pathway and synthesis of MR1 ligands associated with the activation of MAIT cells. Their findings indicate that when either Msm or Mtb lacks key enzymes involved in the early or intermediate stages of this pathway, there is a significant reduction in MAIT cell activation. The use of radiolabeled peptides from the RibA2, RibC, and RibH genes were a rigorous and complementary analysis to supplement the gene expression data. The authors show that deletion of RibA2 or RibG resulted in 82-99% reduction in MR1T cell activation, but RibC and RibH deletions did not. Yet RibH deletions did reduce activation of a TRAV1-2- clone, which uniquely recognized Msm (not Mtb), implying it was lumazine-specific. Additionally, the study highlights that the impact on MAIT cell activation differs based on the specific mycobacterial strain involved, particularly when there is a deficiency in the RibH enzyme. Overall, the research underscores a more diverse set of ligands for MR1T cells, and the importance of riboflavin-synthesis intermediates as MR1 ligands for mycobacterial-specific MAIT cell responses. There were only minor issues noted with referencing one of the MR1T clones in the text and in Figure 5. The paper is well-written, clear and straightforward. The model used is well-explained and the use of complemented mutant strains is relevant to their claim.

Reviewer #3: This a study comparing Mycobacterium smegmatis (Msm) M. tuberculosis (Mtb) viability, growth kinetics, gene expression, and immunogenicity after targeted gene deletion of enzymes in the riboflavin biosynthetic pathway. The manuscript is well-written. The topic is significant as new drug targets are needed against Mtb. Moreover, the riboflavin biosynthetic pathway generates immune ligands for MAIT cells restricted by a highly conserved antigen presenting system in mammals, MR1, and holds potential for immune-directed therapeutics. The authors confirm prior findings, including from the authors themselves, that ribA2, ribG, ribH, and ribC are essential genes in Mtb and further validate results of their prior publication that these strains can be rescued with riboflavin supplementation and genetic complementation. The authors also highlight ribH gene redundancy in Msm. The highest impact findings of the manuscript are that (1) deletion of Msm and Mtb genes upstream of the canonical activating MR1 precursor ligand, 5-A-RU, suppresses but does not abrogate MAIT cell IFNg responses and (2) deletion of ribH in Mtb that lies downstream of 5-A-RU, leads to enhanced MAIT cell IFNg responses in a previously published select donor biorepository using in vitro infection models. The authors conclude that 5-A-RU is an essential MAIT cell precursor activating ligand in mycobacteria, paralleling published findings in other bacteria. However, this conclusion was made by inference only, as no direct measurements of metabolite levels were made. We thus recommend either additional experiments or greater caution, as detailed further below. The authors speculate that additional riboflavin-independent MR1 ligands may be generated during Mtb infection to explain why riboflavin auxotrophs remain immunogenic to MAIT cells.

**Part II – Major Issues: Key Experiments Required for Acceptance**

Reviewer #1: 1) General note on data in Fig. 2-6 and associated Supplementary Figures:

- For experiments in Fig. 2, Fig. 3 only 2 biological replicates are shown and SD is plotted. SD can only be plotted for 3 or more biological replicates and experiments need to be done at least 3 times.

- How come Fig S3 says it shows data from 3 biological replicates, but more than 3 dots are seen in each bar and why do the number of dots differ between bars in the same graphs? Also, in Fig S4 it says data is from 3 independent experiments but in the bar chart there are 4 dots in each bar. In Fig. 3, the Figure legend states that there were 3 biological replicates for A, C and E but for the DIA measurements the number of biological replicates was not indicated (although here the number of dots in the bars suggests it has been done 3-times).

- In all cases not only mean values but also values from individual experiments should be shown. This was done for most Figures but please amend this also for Fig. 3 and 4 A, C, E.

- Fig. 2 requires stats.

- In Fig. 3, 4, S3 it looks like stats were only done between mutant+RF versus reconstituted mutant but should also be done between mutant+RF and Msm WT+RF.

- Fig 4 B, D, F –it looks like the data for the condition Mtb WT+RF was omitted from the graph whilst it appears in the legend. Accordingly, I cannot comment on text in lines 210 and 212 wrt protein levels.

- Fig. 5, 6 The absolute IFNg production of each T cell clone to wt Msm and in response to a pan-stimulus like PMA/Ionomycin or anti-CD3/CD28 also needs to be shown.

2) Growth phenotype of mutants and mutants reconstituted with genes (Fig. 2, S2, Fig. 3, Fig. 5):

- Why is there only limited to no complementation observed for the Msm ΔribA2, ΔribH1 and ΔribH2ΔribH1 strains based on growth kinetics (Fig. 2) and transcript and protein of the relevant genes (Fig. 3) as well as T cell activation (Fig. 5 for ribH2, alone or in combination with ribH1)? Was the reconstitution done several times? The reconstitution is an important verification of the mutant and when complementation does not work this casts doubt on the specificity of the mutants.

- Fig. S2 assesses growth phenotyping upon streaking. It is difficult to interpret the plate images given that too many bacteria have been streaked out and individual colonies cannot be identified. The only conclusion that can be drawn from the data presented is of qualitative nature - bacteria grew or not. Instead, the authors should quantify the numbers of colonies and the size of the colonies upon plating a certain volume of bacteria (rather than streaking them out) to determine the growth phenotype. Letting the bacteria grow for a shorter time frame would allow for individual colonies to be identified the size of which can be measured.

- 158: Shouldn’t also the ΔribH2 mutant of Msm be listed amongst those mutants that showed no growth phenotype and the ΔribH1 mutant of Msm amongst those mutants that were auxotrophic for riboflavin?

3) Mutant ko’s generated and their phenotypes

- Why were no mutant ko’s of ribG and fbiC for Mtb generated?

- The hypothesised redundancy of the lumazine synthase activity in Msm should be described consistently and in a more cautious way in lines 142 and 148. Fig 2A shows that in Msm deletion of ribH1 but not of ribH2 showed an impact on growth. In addition Fig 3 shows that albeit an increase in transcript and protein level of ribH2 in the ribH1 mutant growth was still impaired. There was only 30% of growth relative to wt in the presence of RF so this indicates little compensation and not as the authors state, ‘cannot fully compensate’ (line 148). It would hence be better to tone down the statement about the redundancy of ribH1 and ribH2 in line 142 and make this a point in the discussion. In line 238 onwards, how do the authors explain the reduced activation by the reconstituted ΔribH2 mutant of the TRAV1-2- clone, D520-E10 which showed no growth defect in Fig 2 and given the redundancy between ribH1 and ribH2?

- 182: ‘a 3-fold increase in ribH2 transcript and RibH2 protein was observed in the Msm ΔribH1 mutant suggesting a regulatory interdependence of expression between the two lumazine synthases (Fig 3G and H).’ I can see a 3-fold increase in ribH2 transcript but not in protein level in Fig. 3H.

- 187 compares the effects of gene knockouts (this study) versus knockdowns (previous study) and this needs to be described and interpreted more accurately: A knock-out mutant of ribF was not analyzed here so cannot be compared. Silencing of ribA2 suppressed the expression of ribG and a trend can also observed in the knock-out. For ribC there is probably no difference. How do the authors explain that silencing of ribA2 suppressed the expression of ribH1 in Msm but there is a significant increase in ribH1 for the knock-out?

- 203: ‘Additionally, stable isotope-labeled standards were made for peptides derived from RibA2, RibC and RibH’ – Fig S3 only has data for RibC and RibH but why not for RibA2?

4) Capacity of ko mutants to stimulate M. smegmatis and M. tuberculosis -specific MR1-reactive T cell clones (Fig. 5, 6).

- For the TRAV1-2 neg clone D520 E10 the data in Fig. 5 panel D is convincing. However, in panel H activation with the ribH2 from 3 biological replicates ranges from 100% response to 20% of wt+RF, meaning that as it stands it is unclear whether the ko mutant of ribH2 has an impact or not. In the same panel there is a non-significant reduction in the reconstituted fbiC mutant from 90% to less than 25 % which is unexpected but again there is a large range in the data from the 3 experiments. The response by this clone alongside one of the other TRAV1-2+ clones needs to be repeated several times for the ribH2, ribH1+ribH2 and fbiC mutants to really understand the response pattern.

- 319: ‘However, residual MAIT cell activation was observed for Msm ΔribA2, Msm ΔribG and Mtb ΔribA2 thus implicating an alternate source/s of minor mycobacterial MAIT cell agonists distinct from the riboflavin pathway.’ The mean and range should be quantified and statistics done relative to untreated and anti-MR1 blocked condition. Background activation is the attractive alternative hypothesis and should be described.

- 249: ‘The DfbiC mutation had no effect on Msm viability (Fig 2) or recognition by the TRAV1-2+ T cell clones, D426-G11 and D481-C7 (Fig 5). However, there was a trend towards reduced recognition of Msm by the TRAV1-2+ clone, D481-F12, and the TRAV19-1 clone, D510-E10 (Fig 5).’ I can only see a trend for a reduced response for the reconstituted Msm ΔfbiC mutant in panels G and H (which is unexpected), whilst the means for the mutant itself sit at 100% and 80%. Accordingly, 320: ‘That Msm also preferentially produces lumazine antigens is suggested by the observation that the ablation of FbiC impacted the recognition of Msm by the non-TRAV1-2 T cell clone D520 E10, which recognizes Msm and not Mtb.’ Needs to be changed/removed.

- For the D520 E10 clone the authors should also test a double mutant for ribA2 or ribG and ribH1 to determine if the 50% reduced activity observed in each case leads to 0% activation when combined. Is the residual 50% response observed for the ribA2, ribC and ribH1 mutants MR1-depdendent? Additional TRAV1-2 neg T cell clones should be tested to see if the pattern of the D520 E10 clone is generalisable for the response to M. smegmatis.

- I don’t understand the normalisation done. It is essential to also show the raw activation data, as it is possible that the response of the conventional CD8+ D454 H1-2 clone changes for the different mutants.

- It would be important to test to which extent there is MR1-dependent activation observed to Mtb ΔribH.

Reviewer #2: - In line #240 the authors describe D510-E10 as a TRAV1-2- clone, but in line #252 they describe it as a TRAV19-1 clone. In line #332, it is listed as non-TRAV1-2 clone 520-E10. In Figure 5, it is also listed as 520-E10. Are these in fact the same MR1T cell clones? If so, it would be helpful to be consistent about how these are named (either with reference to both Va alleles (TRAV1-2- and TRAV19+) or just reference one for clarity.

- In lines 251-252, the authors state “there was a trend towards reduced recognition of Msm by the TRAC1-2+ clone D481-F12, and the TRAV19-1 clone, D510-E10”. How do the authors explain this effect after fibC deletion in Msm since an antagonist is eliminated and by mass effect, substrate (5-A-RU) should be increased? Is it in fact increased in the fibC mutant? It would enhance this already rigorous dataset and manuscript if the authors could provide these details.

Reviewer #3: The authors speculate that the abundance of metabolic intermediates 5-A-RU or ribityllumazines that are known to comprise classes of activating MR1 ligands can explain MAIT cell IFNg ELISPOT results, however, no experiments are performed to define these metabolites. Profiling riboflavin metabolite levels in the knockout strains is necessary to make any conclusions about ligands responsible for changes in MAIT cell activation.

Insufficient detail is provided about the infection protocols used for the activation assays. Were bacteria used straight from thawed stocks or first cultured, how were CFUs determined to calculate the MOI, and how long did the infection proceed prior to the IFNg release assay? Besides the concern about the metabolic state of just-thawed bacteria, if they were grown with RF supplement, there is presumably also residual intracellular riboflavin that could affect metabolite levels (even if transcript and/or protein levels are unchanged), also depending on the duration of infection (e.g., how long the Mtb were depleted of riboflavin, assuming that there is no access to riboflavin in the host cell). Please provide more detailed and complete methods for the infection to enable better insights into the potential state of the bacteria and resulting presentation/stimulation of host cells.

As it stands, the authors convincingly show that they can generate riboflavin gene knockouts in Msm and Mtb and demonstrate notable distinctions between species, but we have several queries regarding the specificity of knockouts:

1. Whole genome sequencing methods are cited but there is no mention of these data in the manuscript or as supplemental data. We assume these were used to confirm knockouts and complemented strains, and importantly, to exclude off-target effects. Did the sequencing indeed confirm the genetic manipulation (borders of knockout/insertion and complement insertion) and the absence of secondary mutations compared to the parent strain?

2. Why does the strain Msm ΔribH2 ΔribH1:: ribH2 ribH1 not grow in the presence of RF? It is possible that this strain is compromised at a different location in the genome (see above) and affecting another essential pathway. This also affects the conclusions drawn from figure 5 on MAIT cell activation by Msm.

3. From figure 2A, it is clear that the growth defect in Msm is only caused by the loss of ribH1. Deletion of ribH2 has no effect on the growth of Msm in the absence of riboflavin. This underscores an alternative interpretation to the author’s conclusion of “redundancy;” ribH1 may be the only functional lumazine synthase in Msm.

4. Expression of other rib genes is variable compared to wild type. For example: ribA2 expression is lower in ΔribG and ΔribH2 and higher in ΔribH1 and ΔribC. Notably, ribH expression in ΔribA2 strain is higher compared to the wild type, attributed to the presence of a vector-borne promoter by the authors. These results confound interpretation of the presumed metabolite perturbations and attempts to functionally validate strains using immune assays. Given the sensitivity of flux in metabolic pathways, target-specific knockouts that preserve the activity of other genes in the pathway become crucial for drawing meaningful conclusions.

Many of these queries may be obviated by metabolic validation, but the manuscript offers limited insights into how these proteomic perturbations impact riboflavin intermediate chemistry.

Importantly, the immune assays used are reductionist using select donors also used in prior publications, do not include data from donors with negative responses to assess generalizability, and do not include measures of MR1-dependence.

Figure 5/6

The authors use a well-established IFNg ELISPOT and MAIT cell clones from specific donors also used in their prior work. Further details regarding the methods are requested—how were DCs generated, were they autologous for each donor? Why were DCs used for Msm and BEAS2B used for Mtb? Was IFNg the only response that can be detected from these MAIT cells? The IFNg readout seems surprising considering MAIT cells require cytokine stimulation to generate significant amounts of IFNg—is this provided by the APC and how many APCs are required to generate this response? What other parameters of MAIT cell activation were tested? Was flow cytometry performed? Representative flow cytometry plots of each MAIT cell and dendritic cell clone used in these experiments after thawing of vials from the biorepository should be included with gating strategy and viability. Reference to previous work is not sufficient as distinct vials from the biorepository were thawed for the experiments in this manuscript and variability is to be expected. Are dendritic cells/BEAS the only APCs that are effective for this assay? Representative images of the ELISPOT should be included for the reader to appreciate how the results are calculated.

While these donors have previously been used in publications, further details regarding the demographics of the donors including Mtb exposure/IGRA status should be reported.

As the same donors are repeatedly used, questions arise regarding the reproducibility of these results in other donors. Has this assay been performed using MAIT cells from other populations? If the assay is not reproducible in other MAIT cell clones, the negative results should also be displayed. What host factors determine the ability for the MAIT cell clone to respond to Mtb? Is prior exposure necessary for or even suppressive of MAIT cell responses during repeat challenge? Are all Mtb-reactive MAIT cells CD8+? One TRAV1-2 negative MR1T was used—what proportion of TRAV1-2 negative MAIT cells are Mtb-reactive? The authors reference their previous work on how these MR1T were expanded, however, a brief summary of the methods should be included for completeness including gating strategy for isolation of the original clones (MR1 tetramer vs anti-TRAV1-2?) and method of expansion (antigen vs. mitogen), cryopreservation methods, and details of culturing after thaw prior to performing ELISPOT (eg, can the clones be thawed and used the same day, how many days are they cultured after initial expansion and prior to cryopreservation, or after thaw). No positive control for antigen-specific activation (eg, 5-OP-RU) of these MAIT cell clones are included to confirm reproducibility. No MR1-dependence is measured (eg, anti-MR1). The authors only include mitogen as a positive control.

Why is the response to Msm mutants measured relative to WT while for Mtb mutants relative to Mtb8.4 T cells? Where is the relative response to Mtb WT?

**Part III – Minor Issues: Editorial and Data Presentation Modifications**

Reviewer #1: - Fig. S1: Shouldn’t ribH2 be indicated here for Msm?

- Fig. 6 Figure legend states the use of DCs as APCs whilst the main text mentions the BEAS-2B bronchial epithelial cell line

- The display of Fig 3, 4, 5, 6 and S3 needs to be improved for clarity: same colour coding for transcript and protein levels in Fig 3, 4. Having x-axis labels instead of a legend would make it easier to digest the data in Fig 3, 4, 5, 6, S3.

- 45/46 and throughout 5-A-RU vs 5-AR-U, 5-A-RU is usually used

- 67 onwards inaccurate referencing: ‘MR1-restricted T cells (MR1Ts) are characterized by their dependence on the highly conserved molecule, MR1 [1].’ Reference 1 is not adequate for MR1 T cells. MR1 restriction was demonstrated by Treiner et al Nature 2003. Kjer-Nielsen et al Nature 2012 discovered that MR1 presents Vitamin B metabolites but 5-OP-RU as MAIT cell antigen was described in Corbett et al Nature 2014.

- 88: references for Salmonella Typhimurium are 7 and 11; for E. coli there is also Soudais et al JI 2015; lastly Lopez-Rodriguez et al JEM 2023 generated a Klebsiella ribD mutant

- 110: ‘in the field of MR-1 (suggesting inserting dependent) immune responses towards Mtb.’

- 123: ‘myriad (insert: of) enzymes’

- Fig 1B is not really touched on in the main text and not properly described in the Figure legend. It would be of interest to describe the operon organisation and regulation, point out what is not known where applicable and also talk about how the operon organisation and regulation is different to other bacteria.

- 145: ‘In prior work, we used a panel of…’ check sentence

- 194: ‘Quantification of riboflavin pathway transcript (insert: ‘and’) protein levels in Msm strains.’

- 195: explain sigA

- 201: cite Fig. 2

- 206: ‘Riboflavin supplement suppressed expression of ribC and enhanced expression of ribA2 in wildtype Mtb (Fig 4A and E) but had no significant impact on expression of other pathway genes (instead: ‘had no impact on expression of ribH’ as only ribH was tested other than ribA2 and ribC).’

Reviewer #2: - Line 41: Name of the enzyme for RibH should be listed here as this is the first use of the abbreviation, instead of in the following line 42

- Fig.3 Figure color-coded legend is confusing as one giant list. It could be moved to the right of the graphs, or upper right corner for each section to improve reading.

- Line #60 authors just mentioned enzyme synthase but did not provide an acronym for LS from Mtb. (Do not consider this super relevant)

- In line 287, RibA2 is listed as both “early” and “penultimate” steps in the riboflavin pathway. Could the authors clarify?

- Line #268-269 “[…] normalized these responses to the HLA-B45, Mtb8.4-specific clone.” -> Line #278-279 explains the normalization of results by the Mtb8.4-specific CD8 T cell clone. This information could append the sentence initiated in #268 for better understanding.

- #320-321 “[…] implicating an alternative source/s of minor mycobacterial MAIT cell agonists […]”. Are there any known alternative sources? Supplementary information on the topic would enrich the discussion.

- #232 typo; “D481 C7” -> D481-C7

- #332 typo; “D525 E10” -> D525-E10

Reviewer #3: The manuscript should include the rationale for using different concentrations of riboflavin supplementation in Msm and Mtb.

The layout of the multicolor/pattern bar graphs and multipart legends in Fig 3-5 can be improved for readability.

Why was Mtb fbiC knockout not included?

Additional discussion is requested as to why ΔribH enhances MAIT cell activation, but not ΔribC when according to the author inference, both should cause accumulation of the activating precursor ligand 5-A-RU.

Line 148: Cited study demonstrates ability to uptake RF, but does not show transport, which generally implies facilitated transport (vs. passive diffusion). Suggest rewording.

Line 155: RibA2 catalyzes formation of a substrate used by RibH, which is the enzyme that catalyzes the penultimate step. Suggest rewording.

Lines 187- 192: The authors are suggesting that the differences in expression of other rib genes between a knockdown and a knockout can be attributed to the presence of residual protein in the knockdown vs no protein in knockout. This implies some mechanism by which target gene expression would affect other genes, but we are not aware of any evidence to this effect. Suggest rewording or otherwise qualifying this statement accordingly.

Line 194: “transcript protein” -> “transcript AND protein”

Line 197: Statistical significance analysis of the mean and S.E.M. with two biological replicates is not widely accepted as rigorous for proteomics. Please clarify this analysis.

Figure 4: The most relevant comparison in gene expression and protein abundance among different strains is to the wild type rather than between the complement and +RF condition. Please include the comparison to the wild type statistical analysis of significance.

Figures 3/5: There is a discrepancy in the smegmatis ribH1/ribH2 KO strains that are characterized for expression/complementation vs. for the immunological assays. In particular the ΔribH1ΔribH2::ribH2 strain in Fig. 5 appears to be used as a ΔribH1 knockout (although this is not explained), but was not validated for RNA/protein expression and so those IFNg release data cannot be interpreted in terms of rib pathway disruption. Suggest that the authors provide fully correspondent data, as is the case for other strains.

PLOS authors have the option to publish the peer review history of their article (what does this mean? ). If published, this will include your full peer review and any attached files.

**Do you want your identity to be public for this peer review?** For information about this choice, including consent withdrawal, please see our Privacy Policy .

Reviewer #1: No

Reviewer #2: **Yes: ** Stephen M. Carpenter

Reviewer #3: No

**Figure resubmission:****Reproducibility:** To enhance the reproducibility of your results, we recommend that authors of applicable studies deposit laboratory protocols in protocols.io, where a protocol can be assigned its own identifier (DOI) such that it can be cited independently in the future. Additionally, PLOS ONE offers an option to publish peer-reviewed clinical study protocols. Read more information on sharing protocols at https://plos.org/protocols?utm_medium=editorial-email&utm_source=authorletters&utm_campaign=protocols

---

## [Decision Letter · Decision Letter 1]

10 Jun 2025-06-10

PPATHOGENS-D-24-02117R1

Disruption of riboflavin biosynthesis in mycobacteria establishes 5-amino-6-D-ribitylaminouracil (5-A-RU) as key precursor of MAIT cell agonists

PLOS Pathogens

Dear Dr. Mizrahi,

Thank you for submitting your manuscript to PLOS Pathogens. After careful consideration, we feel that it has merit but does not fully meet PLOS Pathogens's publication criteria as it currently stands. Therefore, we invite you to submit a revised version of the manuscript that addresses the points raised during the review process.

Please submit your revised manuscript within 30 days Aug 09 2025 11:59PM. If you will need more time than this to complete your revisions, please reply to this message or contact the journal office at plospathogens@plos.org. Please include the following items when submitting your revised manuscript:

We look forward to receiving your revised manuscript.

Kind regards,

Alice Prince

Section Editor

PLOS Pathogens

Sumita Bhaduri-McIntosh

Editor-in-Chief

PLOS Pathogens

orcid.org/0000-0003-2946-9497

Michael Malim

Editor-in-Chief

PLOS Pathogens

orcid.org/0000-0002-7699-2064

**Additional Editor Comments :**

As noted below - the reviewers are enthusiastic about this work and now have only a few small issues that need to be addressed.

**Reviewers' Comments:**

Reviewer's Responses to Questions

**Part I - Summary**

Reviewer #1: I commend the authors on the thoroughness of their rebuttal. There were however 2 queries that hadn't been addressed, outlined below.

Reviewer #2: This paper reveals a diverse set of ligands for MR1-restricted T cells, and emphasizes the importance of riboflavin-synthesis intermediates as MR1 ligands for mycobacterial-specific MAIT cell responses. The paper is well-written, and revisions were detailed and responsive to reviewer feedback. The model used is well-explained and the use of complemented mutant strains is rigorous despite technical challenges. All major issues were addressed in revision. Only minor typographical edits are suggested in this revised manuscript.

Reviewer #3: The authors have addressed most of the reviewer comments, revised figures to more clearly compare strains, revised immunology data to exclude clones with inconsistent results and now include detailed methods. Major additions are metabolomic outcomes.

1. We still think the authors cannot conclude that 5-A-RU is the central MAIT cell activator without directly measuring it and a revised title could more cautiously state "riboflavin precursor intermediates" as they do not know it is 5-A-RU vs. some other undiscovered metabolite.

2. A remaining inconsistency pertains to Msm ribH1/ribH2 data. The data on solid medium supports redundancy of ribH1 and ribH2 in Msm (Fig. S2). However, the authors should comment on discrepancies in the liquid culture data that are not addressed by their qPCR results on ribH1, which is only for the dKO double complement. The unaddressed discrepancies are as follows:

a. ribH2 KO complement grows less well than the KO, which at least as shown has a minor growth phenotype. Do the authors assume that these differences vs. WT are not significant? Do they have analysis to show that these differences are not significant while others (e.g., in Fig. 2D and 2E for other ribH mutants) are, to better support their conclusions?

b. The ribH1 KO complement does not complement at all (Fig. 2D); this is not addressed by their qPCR data, which is only for the dKO double complement in Fig. 2E.

Ultimately, the authors wish to emphasize that these strains are indistinguishable, but the microbiological discrepancies should at least be acknowledged and discussed.

Lines 325-330: While it is fine to confirm by MS that riboflavin is still present in all strains, including auxotrophs grown on riboflavin as shown in Fig. S8A, C, the authors should be more clear that when they are discussing the ribH mutants in this section, these strains were grown WITHOUT riboflavin (Fig. S8C).

3. An additional minor point is that the off target effects of mutants on other proteins is not detailed in the text (lines 216-220, eg, Fig 3).

4. Finally, in Supp Fig 8 the authors state that metabolomic comparisons were only made in conditions not spiked with riboflavin, but do not explicitly state this for other metabolomic analyses (eg Fig 6, figure legend is missing). Riboflavin supplementation could affect metabolomic results and the preparation conditions for MS should be clarified.

Overall the manuscript is greatly improved from initial submission.

**Part II – Major Issues: Key Experiments Required for Acceptance**

Reviewer #1: NA

Reviewer #2: All major concerns have been addressed.

Reviewer #3: (No Response)

**Part III – Minor Issues: Editorial and Data Presentation Modifications**

Reviewer #1: I copy paste here the 2 items from the rebuttal, including my initial query, the response and the follow-up query:

3) Fig. 2 requires stats.

Response: Corrected. Statistical comparisons, against wildtype were performed using

a one-way ANOVA and Dunnett’s multiple comparison test.

Follow-up: The resolution of the Figure is low and I cannot see one or two stars for stats in the Figure as per the legend. The authors seem to have added a small extra legend in some of the plots which might be stats but it is unclear which time point they refer to. If the stats are done for the 12 hr timepoint, please clarify this in the legend.

17) 319: ‘However, residual MAIT cell activation was observed for Msm ΔribA2, Msm

ΔribG and Mtb ΔribA2 thus implicating an alternate source/s of minor mycobacterial

MAIT cell agonists distinct from the riboflavin pathway.’ The mean and range should

be quantified and statistics done relative to untreated and anti-MR1 blocked condition.

Background activation is the attractive alternative hypothesis and should be described.

Response: To address this concern, we have now used an anti-MR1 antibody to

demonstrate that the residual activity in the ribA2 and ribG gene deletion strains is

MR1-dependent, shown in S11 Fig. These data would suggest that not all ligands are

5-A-RU dependent, and we have added more granularity to the discussion to reflect

this. We can say confidently that these T cell clones are MR1 restricted in that they

bind the MR1/5-OP-RU tetramer and do not respond to MR1 knockout cell lines

pulsed with Msm or Mtb, and respond to both canonical and non-canonical MR1

antigens (Gold et. Al, PLOS Biology (2010), Harriff et al, Sci Immunol (2018);

Narayanan et al, Sci. Rep (2020), Culicke et al, Commun. Biol. (2024)). Hence, we

have not observed bystander activation in these clones.

Follow-up: It is great to see the MR1 blocking data in Fig S11 and S13. The stats of each mutant relative to untreated and to the anti-MR1 blocked condition are still missing. If there is no significant increase in untreated vs the rib ko treated, then the conclusion is that the activation observed is background. If there is a significant increase in activation that is then significantly blocked by anti-MR1 antibody, then the conclusion is as currently stated: alternate source/s of minor mycobacterial MAIT cell agonists distinct from the riboflavin pathway. The latter should further be clarified as to wether this was observed in all or selected clones. Any statements highlighting the existence and importance of alternate sources need to be adapted based on these results.

I think there might be a chance that there is residual significant activation in the case of Msm ribA2 and ribG mutants in Fig S11.

However, in Fig S13, there now is no response in the context of the ribA2 mutant of Mtb by any of the 3 clones tested, so that there are no riboflavin independent antigens. Accordingly the sentence in the discussion line 512 needs to be changed:

However, residual MAIT cell activation that was MR1-dependent was observed for Msm DribA2, Msm DribG and Mtb DribA2 thus implicating an alternate source/(s) of minor mycobacterial MAIT cell agonists distinct from the riboflavin pathway.

Reviewer #2: Two minor concerns with revised text are listed here:

- Line 410 should read “two of the MR1-restricted T cell clones” (instead of three) since only two are listed.

- Line 422: “which was most notably” -> I believe should be “notable”

Reviewer #3: (No Response)

PLOS authors have the option to publish the peer review history of their article (what does this mean? ). If published, this will include your full peer review and any attached files.

**Do you want your identity to be public for this peer review?** For information about this choice, including consent withdrawal, please see our Privacy Policy .

Reviewer #1: No

Reviewer #2: **Yes: ** Stephen M. Carpenter

Reviewer #3: No

**Figure resubmission:**
---

## [Editor Report · Decision Letter 2]

19 Jun 2025

Dear Mizrahi,

We are pleased to inform you that your manuscript 'Disruption of riboflavin biosynthesis in mycobacteria establishes riboflavin pathway intermediates as key precursors of MAIT cell agonists' has been provisionally accepted for publication in PLOS Pathogens.

Best regards,

Alice Prince

Section Editor

PLOS Pathogens

Alice Prince

Section Editor

PLOS Pathogens

Sumita Bhaduri-McIntosh

Editor-in-Chief

PLOS Pathogens

orcid.org/0000-0003-2946-9497

Michael Malim

Editor-in-Chief

PLOS Pathogens

orcid.org/0000-0002-7699-2064

These revised manuscript is an excellent contribution to this topic and will be of substantial interest to the field.
---

## [Editor Report · Acceptance letter]

Dear Mizrahi,

We are delighted to inform you that your manuscript, "Disruption of riboflavin biosynthesis in mycobacteria establishes riboflavin pathway intermediates as key precursors of MAIT cell agonists," has been formally accepted for publication in PLOS Pathogens.

Best regards,

Sumita Bhaduri-McIntosh

Editor-in-Chief

PLOS Pathogens

orcid.org/0000-0003-2946-9497

Michael Malim

Editor-in-Chief

PLOS Pathogens

orcid.org/0000-0002-7699-2064